# ATTA: Anomaly-aware Test-Time Adaptation for Out-of-Distribution Detection in Segmentation

**Zhitong Gao[1], Shipeng Yan[1], Xuming He[1,2]**
[1]School of Information Science and Technology, ShanghaiTech University
[2]Shanghai Engineering Research Center of Intelligent Vision and Imaging
`{gaozht,yanshp,hexm}@shanghaitech.edu.cn`

## Abstract

Recent advancements in dense out-of-distribution (OOD) detection have primarily focused on scenarios where the training and testing datasets share a similar domain, with the assumption that no domain shift exists between them. However, in real-world situations, domain shift often exits and significantly affects the accuracy of existing out-of-distribution (OOD) detection models. In this work, we propose a dual-level OOD detection framework to handle domain shift and semantic shift jointly. The first level distinguishes whether domain shift exists in the image by leveraging global low-level features, while the second level identifies pixels with semantic shift by utilizing dense high-level feature maps. In this way, we can selectively adapt the model to unseen domains as well as enhance model's capacity in detecting novel classes. We validate the efficacy of our method on several OOD segmentation benchmarks, including those with significant domain shifts and those without, observing consistent performance improvements across various baseline models. Code is available at `https://github.com/gaozhitong/ATTA`.

## 1   Introduction

Semantic segmentation, a fundamental computer vision task, has witnessed remarkable progress thanks to the expressive representations learned by deep neural networks [33]. Despite the advances, most deep models are trained under a close-world assumption, and hence do not possess knowledge of what they do not know, leading to over-confident and inaccurate predictions for the unknown objects [18]. To address this, the task of *dense out-of-distribution (OOD) detection* [1, 15], which aims to generate pixel-wise identification of the unknown objects, has attracted much attention as it plays a vital role in a variety of safety-critical applications such as autonomous driving.

Recent efforts in dense OOD detection have primarily focused on the scenarios where training and testing data share a similar domain, assuming no domain shift (or covariant shift) between them [50, 3, 1, 15, 38]. However, domain shift widely exists in real-world situations [39] and can also be observed in common dense OOD detection benchmarks [29]. In view of this, we investigate the performance of existing dense OOD detection methods under the test setting with domain-shift and observe significant performance degradation in comparison with the setting without domain-shift (cf. Figure 1). In particular, the state-of-the-art detection models typically fail to distinguish the distribution shift in domain and the distribution shift in semantics, and thus tend to predict high uncertainty scores for inlier-class pixels.

A promising strategy to tackle such domain shift is to adapt a model during test (known as test-time adaptation (TTA) [46]), which utilizes unlabeled test data to finetune the model without requiring prior information of test domain. However, applying the existing test-time domain adaption (TTA) techniques [46, 7, 31, 47] to the task of general dense OOD detection faces two critical challenges. First, traditional TTA methods often assume the scenarios where all test data are under domain shift

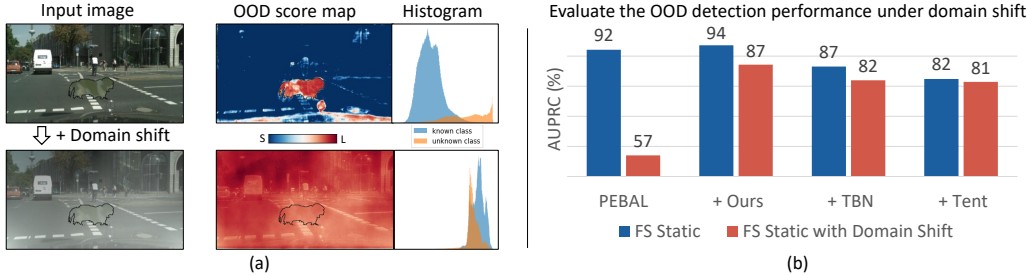

Figure 1: (a) We visualize OOD score maps and the corresponding histograms generated by the SOTA method PEBAL [44] on both an original and a domain-shifted (smog-corrupted) image. (b) We quantify the drop in PEBAL's performance with the added domain shift and compare it to the performance when combined with our method or existing test-time adaptation methods such as TBN [36] and Tent [46]. Please refer to Section 4.3 for additional results.

while our dense OOD detection task addresses a more realistic setting where test data can come from seen or unseen domains without prior knowledge. In such a scenario, TTA techniques like the transductive batch normalization (TBN) [36, 40, 2], which substitutes training batch statistics with those of the test batch, could inadvertently impair OOD detection performance on images from seen domains due to inaccurate normalization parameter estimation (cf. Figure 1(b)). On the other hand, the existence of novel classes in test images further complicates the problem. Unsupervised TTA losses like entropy minimization [46, 12, 32, 45] often indiscriminately reduce the uncertainty or OOD scores of these novel classes, leading to poor OOD detection accuracy (cf. Figure 1(b)). Consequently, how to design an effective test-time adaptation strategy for the general dense OOD detection in wild remains an open problem.

In this work, we aim to address the aforementioned limitations and tackle the problem of dense OOD detection in wild with both domain- and semantic-level distribution shift. To this end, we propose a novel dual-level test-time adaptation framework that simultaneously detects two types of distribution shift and performs online model adaptation in a selective manner. Our core idea is to leverage low-level feature statistics of input image to detect whether domain-level shift exists while utilizing dense semantic representations to identify pixels with semantic-level shift. Based on this dual-level distribution-shift estimation, we design an anomaly-aware self-training procedure to compensate for the potential image-level domain shift and to enhance its novel-class detection capacity based on re-balanced uncertainty minimization of model predictions. Such a selective test-time adaptation strategy allows us to adapt an open-set semantic segmentation model to a new environment with complex distribution shifts.

Specifically, we develop a cascaded modular TTA framework for any pretrained segmentation model with OOD detection head. Our framework consists of two main stages, namely a selective Batch Normalization (BN) stage and an anomaly-aware self-training stage. Given a test image (or batch), we first estimate the probability of domain-shift based on the statistics of the model's BN activations and update the normalization parameters accordingly to incorporate new domain information. Subsequently, our second stage performs an online self-training for the entire segmentation model based on an anomaly-aware entropy loss, which jointly minimizes a re-balanced uncertainty of inlier-class prediction and outlier detection. As the outlier-class labels are unknown, we design a mixture model in the OOD score space to generate the pseudo-labels of pixels for the entropy loss estimation.

We validate the efficacy of our proposed method on several OOD segmentation benchmarks, including those with significant domain shifts and those without, based on FS Static [1], FS Lost&Found [1], RoadAnomaly [29] and SMIYC [3]. The results show that our method consistently improves the performance of dense OOD detection across various baseline models especially on the severe domain shift settings, and achieves new state-of-the-arts performance on the benchmarks.

To summarize, our main contribution is three-folds: (i) We propose the problem of dense OOD detection under domain shift (or covariance shift), revealing the limitations of existing dense OOD detection methods in wild. (ii) We introduce an anomaly-aware test-time adaptation method that jointly tackles domain and semantic shifts. (iii) Our extensive experiments validate our approach, demonstrating significant performance gains on various OOD segmentation benchmarks, especially those with notable domain shifts.

## 2  Related Work

**Dense Out-of-distribution Detection**    The task of Out-of-distribution (OOD) detection aims to identify samples that are not from the same distribution as the training data [18]. While most work focuses on image-level out-of-distribution detection [18, 27, 25, 19, 30, 42], some researchers have begun to study dense OOD detection [1, 15, 3] (or anomaly segmentation), which is a more challenging task due to its requirement for granular, pixel-level detection and the complex spatial relationships in images. Existing work dealing with dense OOD detection often resorts to either designing specialized OOD detection functions [15, 21, 29, 48] or incorporating additional training objectives or data to boost performance in OOD detection [10, 4, 44]. Different from these approaches, our work aims to enhance the model's ability to detect OOD objects at test time by utilizing online test data and design a model-agnostic method applicable to most differentiable OOD functions and training strategies. Furthermore, we target for a more general dense OOD detection problem, where domain-shift potentially exists, which brings new challenges unaddressed in the prior literature.

**Test-Time Domain Adaptation**    The task of test-time domain adaptation (TTA) [46] aims to study the ability of a machine learning model trained on a source domain to generalize to a different, but related, target domain, using online unlabeled test data only. Existing work on TTA mostly tackles the problem from two aspects: adapting standardization statistics in normalization layers and adapting model parameters as self-training. The first line includes utilizing test-time statistics in each batch [36] or moving averages [35, 20], or combining source and target batch statistic [40, 51, 22, 28, 52]. The self-training technique including entropy minimization [12, 32, 45] and self-supervised losses [43, 31]. In our work, we also approach test-time adaptation from these two perspectives. However, our primary distinction lies in addressing a more general open-world scenario, where test data can originate from seen or unseen domains, and novel objects may exist. Consequently, we design our method by explicitly considering these factors and aiming to jointly tackle domain shift and semantic shift challenges.

**Novel Class in Unseen Domain**    While the two primary forms of distribution shift, domain shift and semantic shift, are typically studied independently in literature, there are instances where both are explored in a more complex setting. One such example is *Open Set Domain Adaptation* [37], which investigates the unsupervised domain adaptation problem in scenarios where the target data may include new classes. These novel classes must be rejected during the training process. Another line of research focuses on *zero-shot learning in the presence of domain shift* [49]. In this case, novelty rejection must be performed during the testing phase. A domain generalization method is often employed, which requires data from other domains during training. Some work also discuss the impact of covariant-shift in OOD detection [34]. However, our study diverges from these in several key aspects. First, we study the problem of semantic segmentation where the impacts of domain and semantic shifts are more complex and nuanced compared to the general classification problems. Second, we focus on the situation when no additional data or prior knowledge of the test domain can be obtained during the training.

## 3  Method

### 3.1  Problem Formulation

We first introduce our general dense OOD detection problem setting with potential domain and semantic distribution shift. Formally, we denote an instance of training data as $(x^s, y^s) \sim P_{XY} \in \mathcal{X} \times \mathcal{Y}^d$, where $x^s, y^s$ denotes the input image and corresponding segmentation label, $\mathcal{X} = R^{3 \times d}$ represents the input space (an image with $d$ pixels), $\mathcal{Y} = [1, C]$ is the semantic label space at each pixel, and $P_{XY}$ is the training data distribution. A test data instance is represented as $(x, y) \sim Q_{XY}$ where $Q_{XY}$ is the test data distribution. In the OOD detection problem, we have $Q_{XY} \neq P_{XY}$ and our goal is to identify all pixels where their labels do not belong to the training label space, i.e., $y_i \notin \mathcal{Y}$. In contrast to previous works, we here consider a general problem setting, *dense OOD detection with potential domain shift*, where in addition the input distributions $P_X \neq Q_X$ but with overlaps in their support. Such domain and semantic-class changes leads to a complex data distribution shift, which poses new challenges for the conventional dense OOD detection approaches that are unable to distinguish different shift types.

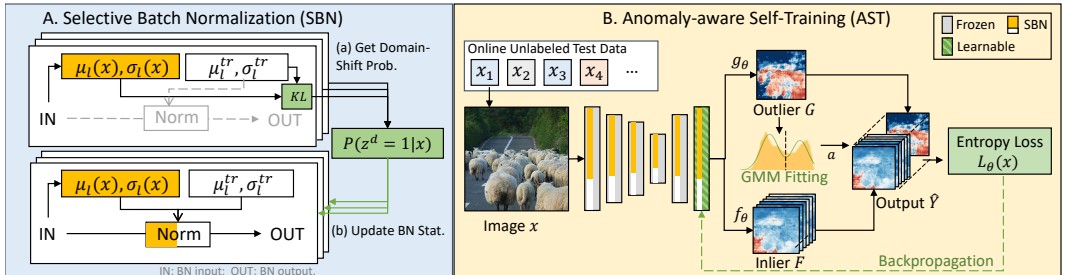

Figure 2: The overview of the two-stage Anomaly-aware Test-Time Adaptation (ATTA) framework. For each test image (or batch), our first stage determines the existence of domain shift and develop a selective Batch-Normalization module to compensate for the input distribution deviation. In the second stage, we devise an anomaly-aware self-training procedure via minimizing a re-balanced uncertainty of model predictions to enhance the OOD detection capacity.

## 3.2   Model Overview

In this work, we aim to address the general dense OOD detection problem by leveraging unlabeled test data and adapting a segmentation model during test time. Specifically, we begin with a segmentation network that has been trained using data from $P_{XY}$. Here we consider a common differentiable OOD network design [5, 30, 15, 1] that consists of two main components: a seen-class classifier $f_\theta : \mathcal{X} \to \mathcal{Y}^d$ and an unseen class detector $g_\theta : \mathcal{X} \to R^d$. Typically, both network components share the same set of parameters $\theta$. For instance, $g_\theta$ can be the negative energy score as in [13, 30, 23] or the maximized logit score as in [15]. During testing, an image (or a batch) sequentially arrives from $Q_{XY}$, and our goal is to update the model parameters for each batch and produce anomaly-aware semantic segmentation outputs.

In order to cope with potential domain and semantic shift in test data, we introduce a novel dual-level test-time adaptation framework, which allows us to simultaneously identify two types of distribution shifts and performs an online self-supervised learning in a selective fashion. Specifically, we instantiate our TTA strategy as a two-stage cascaded learning process. Our first stage determines the existence of domain shift in the image by exploiting the consistent changes in image-level feature statistics. Based on the inferred shift probability, we develop a selective Batch-Normalization module to compensate for the input distribution deviation. In the second stage, we devise an anomaly-aware self-training procedure to enhance the model's capability in detecting novel classes. The procedure iteratively refines the outlier estimation based on a mixture model of OOD scores and minimizes a re-balanced uncertainty of pixel-wise model predictions for effective adaptation to the test data. An overview of our framework is demonstrated in Figure 2.

We update the model parameters in an episodic manner, where for each (batch of) image(s) we use a same initial network parameter. This makes it more resilience if there is no apparent connection between two different (batches of) image(s). In the following, we first explain our design of the selective BN module in Sec. 3.3, followed by our anomaly-aware self-training stage in Sec. 3.4.

## 3.3   Selective Test-Time Batch Normalization

Our first-stage module aims to identify whether an input image is from the seen or unseen domain and perform model adaptation to compensate any potential domain shift. To achieve this, we take inspiration from the transductive Batch-Normalization (TBN) [36], which is a common strategy of exploiting test-time data for model adaptation. The vanilla TBN, however, often suffers from unstable estimation due to the small size of test data, and hence can lead to performance degradation for our general setting where test data may have varying levels of domain shift.

To tackle this, we introduce a selective BN module that first estimates the probability of input image being generated from unseen domain, and then performs a mixture of Conventional batch normalization (CBN) and TBN according to the inferred domain-shift probability. Specifically, we denote $P(z^d = 1|x)$ as the probability of an image $x$ from an unknown domain, and we estimate the probability by considering the distribution distance between the deep features of test and training data in the Normalization layers of the segmentation network. Formally, let $\mu_l^{tr}, \sigma_l^{tr}$ be the running mean

and standard deviation at the $l$-th BN layer calculated in the end of the model training, $\mu_l(x), \sigma_l(x)$ be the mean and standard deviation at the $l$-th BN layer calculated for each test input $x$, we compute the domain-shift probability as follows:

$$P(z^d = 1|x) = h_{a,b}\left(\sum_{l=1}^{L}\left(KL(\mathcal{N}(\mu_l(x), \sigma_l(x))||\mathcal{N}(\mu_l^{tr}, \sigma_l^{tr}))\right)\right), \tag{1}$$

where $\mathcal{N}$ denotes the normal distribution, $KL$ denotes the Kullback–Leibler divergence, and the $h_{a,b}(x) = \text{sigmoid}((x + a)/b)$ is a sigmoid function with linear transform, which normalizes the distance into a probability value. The parameters $a, b$ are estimated based on the variance of training data statistics such that a data point that is far away from the training statistics has higher probability of image-level domain shift. We then update the BN statistics of the network according to the above probability as follows:

$$\hat{\mu}_l = P(z^d = 1|x) * \mu_l(x) + P(z^d = 0|x) * \mu_l^{tr}, \tag{2}$$

$$\hat{\sigma}_l^2 = P(z^d = 1|x) * \sigma_l^2(x) + P(z^d = 0|x) * (\sigma_l^{tr})^2, \tag{3}$$

where $l \in [1, L]$ and $L$ is the max depth of the BN layers. Such an adaptive BN enables us to balance between a stable BN for the data without domain-shift and a test-time BN tailored to the data with domain shift.

### 3.4 Anomaly-aware Self-training Procedure

After compensating for the potential domain gap, we introduce a second stage of test-time model adaptation, aiming for enhancing the model capacity in OOD detection and closed-set prediction on the test data. To this end, we propose an online self-training procedure for the entire segmentation model based on an anomaly-aware prediction entropy loss. By minimizing a weighted uncertainty of inlier-class prediction and outlier detection, we are able to promote the confidence of model predictions, which leads to a more discriminative pixel-wise classifier on the test data.

Formally, we construct a $(C + 1)$-class pixelwise probabilistic classifier based on the two network modules $f_\theta$ and $g_\theta$ and denote its output distribution as $\hat{Y} \in [0, 1]^{(C+1) \times d}$ with each element $\hat{Y}_{c,i}$ being the probability of $P_\theta(y_i = c|x)$, $c \in [1, C + 1]$. For each pixel $i$, the first $C$ channels represent the probabilities of being each closed-set class while the last channel is the probability of being an outlier class. Given the output distribution, we define our learning objective function as,

$$\mathcal{L}_\theta(x) = -\sum_i \sum_{c=1}^{C+1} w_c \hat{Y}_{c,i} \log(\hat{Y}_{c,i}), \tag{4}$$

where $w_c$ is the weight for class $c$. We introduce the class weights to cope with the class imbalance problem between the inlier and outlier classes as the latter typically have much lower proportion in images. More specifically, we set $w_c = 1$ for $1 \leq c \leq C$ and $w_c = \lambda > 1$ for $c = C + 1$. In the following, we first describe how we estimate the output probabilities $\hat{Y}$ based on the seen-class classifier $f_\theta$ and the OOD detector $g_\theta$, followed by the loss optimization procedure.

**Anomaly-aware output representation** To derive the output probability $\hat{Y}$, we introduce an auxiliary variable $Z^o \in \{0, 1\}^d$, where $Z_i^o = 1$ indicates the $i$-th pixel is an outlier, and $P_\theta(Z_i^o = 1|x)$ denotes the corresponding probability. We also assume that the seen classifier $f_\theta$ outputs a pixelwise probability map $F \in [0, 1]^{C \times d}$ for the known classes $c \in \mathcal{Y}$ and the OOD detector $g_\theta$ generates a pixelwise outlier score map $G \in \mathbb{R}^d$. The anomaly-aware output distribution $\hat{Y}_i$ for the $i$-th pixel can be written as,

$$\hat{Y}_{c,i} = P_\theta(\hat{y}_i = c|x, Z_i^o = 0)P_\theta(Z_i^o = 0|x) + P_\theta(\hat{y}_i = c|x, Z_i^o = 1)P_\theta(Z_i^o = 1|x) \tag{5}$$

$$= F_{c,i}(1 - P_\theta(Z_i^o = 1|x))[\![c \in \mathcal{Y}]\!] + P_\theta(Z_i^o = 1|x)[\![c = C + 1]\!], \tag{6}$$

where $[\![\cdot]\!]$ is the indicator function, and we use the fact that $P_\theta(\hat{y}_i = c|x, Z_i^o = 0) = F_{c,i}$ for $c \in \mathcal{Y}$ and 0 for $c = C + 1$, and $P_\theta(\hat{y}_i = c|x, Z_i^o = 1) = 0$ for $c \in \mathcal{Y}$ and 1 for $c = C + 1$.

Given the marginal probability in Eqn (6), our problem is reduced to estimating the pixelwise outlier probability $P_\theta(Z_i^o = 1|x)$. However, as we only have an arbitrary outlier score map $G$ output by $g_\theta$,

it is non-trivial to convert this unnormalized score map into a valid probability distribution. A naive nonlinear transform using the sigmoid function or normalizing the scores with sample mean could lead to incorrect estimation and hurt the performance. To tackle this problem, we develop a data-driven strategy that exploits the empirical distribution of the pixelwise OOD scores to automatically calibrate the OOD scores into the outlier probability $P_\theta(Z_i^o|x)$.

More specifically, we observe that the empirical distributions of pixelwise OOD scores $\{G_i\}$ appear to be bimodal and its two peaks usually indicate the modes for inlier and outlier pixels (cf. Figure 1). We therefore fit a two-component Mixture of Gaussian distribution in which the components with lower mean indicates the inliers and the one with higher mean corresponds to the outliers. Given the parameters of the Gaussian components, we now fit a sample-dependent Platt scaling function to estimate the outlier probability as follows,

$$P_\theta(Z_i^o = 1|x) = \text{sigmoid}((G_i - a(x))/b(x)), \tag{7}$$

where we set the calibration parameter $a(x)$ as the value achieving equal probability under two Gaussian distributions, i.e., $a(x) = \max\{a : \pi_1 N(a|\mu_1, \sigma_1) = \pi_2 N(a|\mu_2, \sigma_2)\}$ where $\pi_1, \pi_2, \mu_1, \mu_2, \sigma_1, \sigma_2$ are the parameters of the GMM model, and $b(x)$ as the standard derivation of the sample OOD scores. We note that while it is possible to analytically compute the outlier probability based on the estimated GMM, we find the above approximation works well in practice and tends to be less sensitive to the noisy estimation.

**Entropy minimization**    After plugging in the estimated $\hat{Y}$, we can re-write our learning objective in Eqn (4) as the following form:

$$\mathcal{L}_\theta(x) = -\sum_i \left( \sum_{c=1}^C \hat{Y}_{c,i} \log(\hat{Y}_{c,i}) + \lambda \cdot \hat{Y}_{C+1,i} \log(\hat{Y}_{C+1,i}) \right) \tag{8}$$

$$= -\sum_i \left( \sum_{c=1}^C F_{c,i}(1 - \bar{G}_i) \log(F_{c,i}(1 - \bar{G}_i)) + \lambda \cdot \bar{G}_i \log \bar{G}_i \right), \tag{9}$$

where $\bar{G}_i = P_\theta(Z_i^o = 1|x)$ denotes the calibrated outlier probabilities. Direct optimizing this loss function, however, turns out to be challenging due to the potential noisy estimation of the outlier probabilities. To mitigate the impact of pixels with unreliable estimates, we select a subset of pixels with high confidence in outlier estimation and adopt a pseudo-labeling strategy to optimize the loss function in an iterative manner. Concretely, in each iteration, we first compute the calibration parameters in Eqn (7) and choose a subregion $D$ based on thresholding the outlier probabilities. Given $a(x), b(x)$ and $D$, we then use the binarized outlier probabilities as the pseudo labels for the loss minimization as follows:

$$\mathcal{L}_\theta(x) \approx -\sum_{i \in D} \left( \sum_{c=1}^C F_{c,i}(1 - t_i) \log(F_{c,i}(1 - \bar{G}_i)) + \lambda \cdot t_i \log \bar{G}_i \right) \tag{10}$$

$$D = \{i : \bar{G}_i < \tau_1 \text{ or } \bar{G}_i > \tau_2\}; \quad t_i = 0 \cdot [\![\bar{G}_i < \tau_1]\!] + 1 \cdot [\![\bar{G}_i > \tau_2]\!], \tag{11}$$

where $\tau_1, \tau_2$ are the threshold parameters. In addition, we set $\lambda = \sum_i [\![t_i = 0]\!] / \sum_i [\![t_i = 1]\!]$ in an image-dependent manner to automatically determine the class weight. Following the common practice in test-time adaptation [46, 7, 31, 47], we only update a subset of network parameters to avoid over-fitting. In this work, we choose to update the final classification block, which brings the benefit of faster inference after each update iteration.

## 4   Experiments

We evaluate our method on several OOD segmentation benchmarks, including those with significant domain shifts and those without, based on FS Static [1], FS Lost&Found [1], RoadAnomaly [29] and SMIYC [3]. Below, we first introduce dataset information in Sec. 4.1 and experiment setup in Sec. 4.2. Then we present our experimental results in Sec. 4.3, 4.4, 4.5.

### 4.1   Datasets

Following the literature [44, 10], we use the Cityscapes dataset [9] for training and perform OOD detection tests on several different test sets, all of which include novel classes beyond the original

Table 1: We benchmark OOD detection methods on the corrupted FS Static dataset (gray rows) and compare with the results on the original dataset (white rows). Our ATTA method improves the model's robustness against corruption when combined with PEBAL.

|  | MSP [17] | Entropy [24] | Max logit [15] | Energy [30] | Meta-OOD [4] | PEBAL [44] | + Ours | + TBN [36] | + Tent [46] |
|---|---|---|---|---|---|---|---|---|---|
| AUC ↑ | 92.36 | 93.14 | 95.66 | 95.90 | 97.56 | 99.61 | **99.66** | 99.25 | 99.04 |
|  | 70.85 | 71.23 | 74.13 | 74.02 | 78.34 | 67.63 | **99.21** | 98.96 | 98.93 |
| AP ↑ | 19.09 | 26.77 | 38.64 | 41.68 | 72.91 | 92.08 | **93.61** | 86.51 | 82.38 |
|  | 10.52 | 14.32 | 23.60 | 22.36 | 52.31 | 57.02 | **87.14** | 81.97 | 81.42 |
| FPR$_{95}$ ↓ | 23.99 | 23.31 | 18.26 | 17.78 | 13.57 | 1.52 | **1.15** | 2.33 | 4.09 |
|  | 100.0 | 100.00 | 89.94 | 89.94 | 100.0 | 97.17 | **2.94** | 4.26 | 4.43 |

Cityscapes labels. We note that these datasets may exhibit varying degrees of domain shift due to their source of construction. In the following, we provide a detailed overview of each dataset.

**The Road Anomaly dataset [29]** comprises real-world road anomalies observed from vehicles. Sourced from the Internet, the dataset consists of 60 images exhibiting unexpected elements such as animals, rocks, cones, and obstacles on the road. Given the wide range of driving circumstances it encapsulates, including diverse scales of anomalous objects and adverse road conditions, this dataset presents a considerable challenge and potential for domain shift.

**The Fishyscapes benchmark [1]** encompasses two datasets: Fishyscapes Lost & Found (FS L&F) and Fishyscapes Static (FS Static). FS L&F comprises urban images featuring 37 types of unexpected road obstacles and shares the same setup as Cityscapes [38], thereby rendering the domain shift in this dataset relatively minimal. On the other hand, FS Static is constructed based on the Cityscapes validation set [9] with anomalous objects extracted from PASCAL VOC [11] integrated using blending techniques. Consequently, this dataset exhibits no domain shift. For both datasets, we first employ their public validation sets, which consist of 100 images for FS L&F and 30 images for FS Static. Then, we test our method on the online test dataset.

**The FS Static -C dataset** is employed to investigate the impact of domain shift on existing OOD detection methods. We modify the original public Fishyscapes Static dataset [1] by introducing random smog, color shifting, and Gaussian blur [16], mirroring the domain shift conditions [8].

**The SMIYC benchmark [3]** consists of two datasets, both encompassing a variety of domain shifts. The RoadAnomaly21 dataset contains 100 web images and serves as an extension to the original RoadAnomaly dataset [29], representing a broad array of environments. On the other hand, the RoadObstacle21 dataset specifically focuses on obstacles in the road and comprises 372 images, incorporating variations in road surfaces, lighting, and weather conditions.

## 4.2 Experiment Setup

**Baselines:** We compare our method with several dense OOD detection algorithms [15, 17, 24, 26, 30, 4, 10, 14, 44] and test-time adaptation algorithms [36, 46]. To evaluate its generalize ability, we implement our method across different OOD detection backbones, including Max Logit [15], Energy [30] and PEBAL [44]. This allows us to examine its performance with varying capacities of OOD detection baselines and different OOD function forms.

**Performance Measure:** Following [4, 10, 14, 44], we employ three metrics for evaluation: Area Under the Receiver Operating Characteristics curve (AUROC), Average Precision (AP), and the False Positive Rate at a True Positive Rate of 95% (FPR95).

**Implementation Details** For a fair comparison, we follow previous work [1, 4, 44] to use DeepLabv3+ [6] with WideResNet38 trained by Nvidia as the backbone of our segmentation models. In our method, the confidence thresholds $\tau_1$ and $\tau_2$ are set to 0.3 and 0.6 respectively. Considering the standard practice in segmentation problem inferences, we anticipate the arrival of one image at a time (batch size = 1). We employ the Adam optimizer with a learning rate of 1e-4. For efficiency, we only conduct one iteration update for each image. The hyperparameters are selected via the FS Static -C dataset and are held constant across all other datasets. See Appendix B for other details.

Table 2: We compare our method on the OOD detection benchmarks: Road Anomaly dataset, Fishyscapes Lost & Found dataset, and Fishyscapes Static dataset. Our method consistently improve upon several OOD detection methods, with particularly significant improvements observed on the Road Anomaly dataset where domain shift is prominent.

| Methods | OoD Data | Road Anomaly AUC ↑ | AP ↑ | FPR$_{95}$ ↓ | FS LostAndFound AUC ↑ | AP ↑ | FPR$_{95}$ ↓ | FS Static AUC ↑ | AP ↑ | FPR$_{95}$ ↓ |
|---|---|---|---|---|---|---|---|---|---|---|
| MSP [17] | ✗ | 67.53 | 15.72 | 71.38 | 89.29 | 4.59 | 40.59 | 92.36 | 19.09 | 23.99 |
| Entropy [24] | ✗ | 68.80 | 16.97 | 71.10 | 90.82 | 10.36 | 40.34 | 93.14 | 26.77 | 23.31 |
| Mahalanobis [26] | ✗ | 62.85 | 14.37 | 81.09 | 96.75 | 56.57 | 11.24 | 96.76 | 27.37 | 11.7 |
| Meta-OoD [4] | ✓ | - | - | - | 93.06 | 41.31 | 37.69 | 97.56 | 72.91 | 13.57 |
| Synboost [10] | ✓ | 81.91 | 38.21 | 64.75 | 96.21 | 60.58 | 31.02 | 95.87 | 66.44 | 25.59 |
| DenseHybrid [14] | ✓ | - | - | - | 99.01 | 69.79 | 5.09 | 99.07 | 76.23 | 4.17 |
| Max Logit [15] | ✗ | 72.78 | 18.98 | 70.48 | 93.41 | 14.59 | 42.21 | 95.66 | 38.64 | 18.26 |
| + ATTA (Ours) | - | **76.60** | **23.96** | **63.49** | **93.53** | **17.39** | **40.69** | 95.48 | **41.23** | 20.89 |
| Energy [30] | ✗ | 73.35 | 19.54 | 70.17 | 93.72 | 16.05 | 41.78 | 95.90 | 41.68 | 17.78 |
| + ATTA (Ours) | - | **77.41** | **25.27** | **62.57** | 93.30 | **17.47** | 43.32 | **96.0** | **41.84** | **17.63** |
| PEBAL [44] | ✓ | 87.63 | 45.10 | 44.58 | 98.96 | 58.81 | 4.76 | 99.61 | 92.08 | 1.52 |
| + ATTA (Ours) | - | **92.11** | **59.05** | **33.59** | **99.05** | **65.58** | **4.48** | **99.66** | **93.61** | **1.15** |

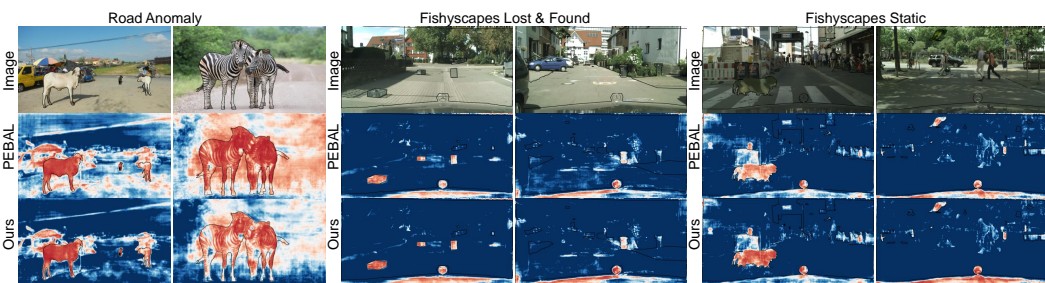

Figure 3: We present qualitative results on the Road Anomaly, FS Lost & Found, and FS Static datasets, where our method improves the previous state-of-the-art model, PEBAL [44], by effectively reducing domain shift and enhancing out-of-distribution detection. This improvement is particularly pronounced in the Road Anomaly dataset, which typically presents a higher domain shift compared to the Cityscapes training set.

### 4.3 Results on simulated FS Static -C Dataset

To investigate the performance of existing OOD detection method in wild, we benchmark several existing dense OOD detection methods [17, 24, 15, 30, 4, 44]. Then we integrate our method with the previous state-of-the-art model on the FS Static dataset, and conduct comparison to test-time adaptation techniques [36, 46].

As shown in Table 1, the introduction of corruption notably affects the performance of OOD detection methods, with an average decrease of 30% in AUC and an average increase of 70% in FPR95. This suggests that current OOD detection methods are highly susceptible to domain shifts. Notably, when combined with our test-time adaptation method, the performance of the state-of-the-art OOD detection method, PEBAL [44], remains more stable in the face of domain shifts, demonstrating a marginal 0.4% drop in AUC. In addition, we note that traditional TTA methods (TBN [36], Tent [46]) can result in performance degradation in the original FS Static dataset where no domain shift occurs (cf. rows in white). By contrast, our method consistently enhances the performance of OOD detection, demonstrating it greater adaptability in real-world scenarios where domain prior information is uncertain.

We present additional results on this dataset in Appendix C, including the results of combining our method with other OOD detection approaches, seen class prediction performance, and experiments under isolated domain shifts.

### 4.4 Results on existing dense OOD detection benchmarks

We then evaluate the performance of our method in existing dense OOD detection benchmarks and integrate it with several OOD detection methods. These include state-of-the-art techniques that do not require retraining or additional OOD data, such as Max Logit [15] and Energy [30], as well as methods that do, such as PEBAL [44]. As shown in Table 2, our ATTA method consistently improve upon previous state-of-the-art models, with particularly notable enhancement observed within the Road Anomaly dataset where the domain shift is more pronounced. Specifically, we enhance PEBAL's performance from 87% to 92% in AUC, 45% to 59% in AP, and reduce the FPR95 error from 44% to 33%. When compared to other methods in the benchmarks, our method in conjunction with PEBAL attains new state-of-the-art performance across all three datasets.

Figure 3 provides qualitative results of our method across these datasets. Our ATTA approach improve the performance of OOD detection backbones by mitigating the impact of domain-shift, and encourage the confidence of the model predictions for both inliers and outliers.

We also submit our method, in combination with PEBAL [44], to various online benchmarks. These include the SegmenMeIfYouCan [3] and the online Fishyscapes [1] benchmarks, where our method consistently outperforms PEBAL [44]. For further details, please refer to Appendix D.3 and D.4.

### 4.5 Ablation Study

We further evaluate the efficacy of our model components on the Road Anomaly dataset using PEBAL [44] as the baseline model. We begin by analyzing the effectiveness of our proposed modules, Selective Batch Normalization (SBN) and Anomaly-aware Self-Training (AST), and subsequently delve into a more detailed examination of each module's design.

As illustrated in Table 3, the individual application of either SBN or AST contributes incremental enhancements to the baseline model shown in the first row. When these two modules are combined, the performance enhancement is further amplified.

Table 3: Ablation study of our two main modules: SBN and AST.

| SBN | AST | AUC ↑ | AP ↑ | $FPR_{95}$ ↓ |
|-----|-----|-------|------|--------------|
| ✗ | ✗ | 87.63 | 45.10 | 44.58 |
| ✗ | ✓ | 88.72 | 48.11 | 43.66 |
| ✓ | ✗ | 90.84 | 55.81 | 37.48 |
| ✓ | ✓ | **92.11** | **59.05** | **33.59** |

We then scrutinized the internal design of each module in Table 4. We first replace SBN with versions that only employ batch-wise statistics (row #1) or training statistics (row #2), demonstrating that our SBN, which incorporates both types of statistics, outperforms these variants. Subsequently, we evaluated the design of our self-training module, initially replacing it with a closed-world entropy [46] calculated only on the seen classes (row #3), and then altered our GMM-based sample-specific normalization to a naive z-score normalization (row #4). We observed that ablating each component resulted in performance degradation, which underscores the effectiveness of our original design.

Table 4: Ablation study of our internal design of each module.

| Train | Batch | Entropy | Norm | AUC ↑ | AP ↑ | $FPR_{95}$ ↓ |
|-------|-------|---------|------|-------|------|--------------|
| ✗ | ✓ | anomaly-aware | GMM | 86.29 | 48.65 | 57.03 |
| ✓ | ✗ | anomaly-aware | GMM | 88.72 | 48.11 | 43.66 |
| ✓ | ✓ | seen-class only | - | 90.46 | 54.64 | 39.28 |
| ✓ | ✓ | anomaly-aware | z-score | 91.25 | 56.65 | 36.33 |
| ✓ | ✓ | anomaly-aware | GMM | **92.11** | **59.05** | **33.59** |

We present the time and memory overhead of the proposed methods, as well as some other ablation results in Appendix E.

# 5   Conclusion

In this work, we propose the problem of dense OOD detection under domain shift, revealing the limitations of existing dense OOD detection methods in wild. To address the problem, we introduce a dual-level OOD detection framework to handle domain shift and semantic shift jointly. Based on our framework, we can selectively adapt the model to unseen domains as well as enhance model's capacity in detecting novel classes. We validate the efficacy of our proposed method on several OOD segmentation benchmarks, including those with significant domain shifts and those without, demonstrating significant performance gains on various OOD segmentation benchmarks, especially those with notable domain shifts.

Despite these promising results, there is still room for further refinement in our ATTA method. Similar to other self-training methods, ATTA relies on a well-initialized model. Our experiments show that our adaptation method yields strong results on state-of-the-art models, but the improvement is less noticeable with weaker backbones. Another practical consideration is the additional time needed for test-time adaptation methods. This is an area where future research could make substantial strides, potentially developing more efficient algorithms that could streamline the process and reduce inference time. All in all, our study introduces a more realistic setting of dense OOD detection under domain shift, and we hope it may offer some guidance or insights for future work in this field.

## Acknowledgments

This work was supported by Shanghai Science and Technology Program 21010502700, Shanghai Frontiers Science Center of Human-centered Artificial Intelligence, and MoE Key Lab of Intelligent Perception and Human-Machine Collaboration (ShanghaiTech University).

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

# Appendix

We provide the pseudo code in Sec. A , more implementation details in Sec. B, additional experimental results in Sec. C and Sec. D, additional ablation results in Sec. E, visualization results in Sec. F and some additional discussion in Sec. G.

## A  Pseudo Code of ATTA

The pseudo code for our proposed method, Anomaly-aware Test-Time Adaptation (ATTA), is presented in Algorithm 1. In this algorithm, each test image $x_t$ utilizes a same pre-trained network parameter $\theta$ for initialization. The adaptation process consists of two stages: a domain-aware selective Batch Normalization (BN) updating stage and an online self-training stage based on an anomaly-aware entropy loss. The final output of the model includes the inlier probability map and outlier score map for each test image.

---

**Algorithm 1:** Anomaly-aware Test-Time Adaptation (ATTA)

---

**Input**  : Test samples $D_{\text{test}} = \{x_t\}_{t=1}^{T}$, seen-class classifier $f_\theta$ and unseen-class detector $g_\theta$ with pretrained $\theta$, number of iterations $N$, learning rate $\eta$, confidence thresholds $\tau_1, \tau_2$.
**Output** : Inlier probability maps $\{F_t\}_{t=1}^{T}$, outlier score maps $\{G_t\}_{t=1}^{T}$.
**for** $x_t \in D_{test}$ **do**
    Initialize parameters $\theta_t \leftarrow \theta$.
    Compute the domain-shift probability based on Eq.(1).
    Update the BN statistics of the network using the above probability as in Eq.(2,3).
    **for** $i = 1, \cdots, N$ **do**
        Get the inlier probability map $F_t \leftarrow f_{\theta_t}(x_t)$ and outlier score map $G_t \leftarrow g_{\theta_t}(x_t)$.
        Calculate the anomaly-aware entropy loss $\mathcal{L}(F_t, G_t, \tau_1, \tau_2)$ as in Eq.(10,11).
        Update the model parameters: $\theta_t \leftarrow \text{Adam}(\nabla\mathcal{L}, \theta_t, \eta)$.
    **end**
    Output the inlier probability map $F_t \leftarrow f_{\theta_t}(x_t)$ and outlier score map $G_t \leftarrow g_{\theta_t}(x_t)$.
**end**

---

## B  More Implementation Details

**Experimental Results of Previous OOD Methods:** All experimental results for the previous dense OOD detection methods in our paper are obtained by running the pretrained model weights provided by the official code repository or as officially reported in their original papers.

**Implementation Details for Tent:** For the Tent method [46], we follow the instructions provided in their original paper regarding their experiments on the segmentation task. This involves considering the parameters of the affine transformation in the Batch Normalization layers as the trainable parameters, using transductive batch normalization [36], and adopting the episodic training manner. To ensure a fair comparison with our method, we update the model only once for each test image, use Adam optimizer and tune the learning rate on the FS Static-C dataset. These steps allow us to align our experimental setup with that of the Tent method and facilitate a meaningful comparison.

**Implementation Details for GMM Modeling in Our Method:** During the process of fitting the Gaussian Mixture Model (GMM) in our method, we employ a peak-finding algorithm to identify the right-most peak in the distribution of OOD scores. By considering this peak as the initial mean value for the right GMM component before performing the Expectation-Maximization (EM) optimization, we aim to mitigate potential issues arising from the presence of multiple peaks in the inlier distribution. This approach leverages the prior knowledge that the right-most peak is more likely to represent the outliers, enhancing the robustness and accuracy of our GMM modeling process.

**Details in Constructing FS Static-C Dataset:** As described in Section 4.1 of our paper, we simulate the domain shift in the FS Static-C dataset by introducing random smog, color shifting, and Gaussian blur. Specifically, we incorporate the fog function from Hendrycks and Dietterich [16] and utilize the 'ColorJitter' and 'GaussianBlur' transformations provided by the torchvision library. Each transformation is independently applied with a 50% probability. By randomly combining these transformations, we generate the final transformed images for the FS Static-C dataset. This process

enables us to create a diverse and challenging dataset that encompasses various forms and levels of domain shift (cf. Sec. F). Consequently, we are able to comprehensively evaluate the performance of our method under different conditions and demonstrate its robustness in handling domain shift.

## C  Additional Results on Simulated FS Static -C Dataset

### C.1  ATTA Combined with Other OOD Detection Methods

We extended our evaluation to include not only PEBAL [44] but also the remaining five OOD detection methods. The results in Table 5 below demonstrate that our method consistently enhances the robustness of these OOD detection techniques in scenarios with potential domain shifts. Specifically, we achieved significant performance gains on the FS Static-C dataset for all OOD detection methods, with an average increase of AUC of around 20% (cf. row #2), AP of 10% (cf. row #4), and a decrease of FPR95 of nearly 80% (cf. last row). Some results on the FS Static-C dataset are even better than the original performance of the corresponding methods on the non-domain-shift FS-Static dataset, such as MSP, Entropy, and Meta-OOD, as shown in the table. By combining our method with various OOD detection methods, we demonstrate its efficacy and general applicability across different scenarios.

Table 5: We display additional results of our method combined with various previously established OOD detection methods on both the FS Static (white rows) and FS Static-C datasets(gray rows).

| | MSP [15] | + Ours | Entropy [17] | + Ours | Max logit [15] | + Ours | Energy [30] | + Ours | Meta-OOD [4] | + Ours |
|---|---|---|---|---|---|---|---|---|---|---|
| AUC ↑ | 92.36 | **93.91** | 93.14 | **95.18** | 95.66 | 95.48 | 95.90 | **96.00** | 97.56 | **98.19** |
| | 70.85 | **92.97** | 71.23 | **94.33** | 74.13 | **94.80** | 74.02 | **95.41** | 78.34 | **98.06** |
| AP ↑ | 19.09 | **26.57** | 26.77 | **39.57** | 38.64 | **41.23** | 41.68 | **41.84** | 72.91 | **83.11** |
| | 10.52 | **20.81** | 14.32 | **30.78** | 23.60 | **31.13** | 22.36 | **32.13** | 52.31 | **75.75** |
| FPR95 ↓ | 23.99 | **20.80** | 23.31 | **18.98** | 18.26 | 20.89 | 17.78 | **17.63** | 13.57 | **11.63** |
| | 100.0 | **22.58** | 100.00 | **20.21** | 89.94 | **23.59** | 89.94 | **18.63** | 100.0 | **11.17** |

### C.2  Segmentation Performance on Seen Classes

In Sec. 4.3 of the main text, we demonstrated the enhancement of OOD detection performance by our method on both the original FS Static dataset and its corrupted version. Here, in Table 6, we further show the improvement of our ATTA method on the segmentation performance of seen classes. We utilize the commonly adopted metrics, mIoU (mean Intersection over Union) and mAcc (mean accuracy), and calculate them exclusively on the inlier pixels.

Notably, our method significantly improves the mIoU metric from 58.83% to 85.64% on the corrupted dataset, approaching the performance level of 87.52% achieved on the original FS Static dataset. This demonstrates the effectiveness of our method not only in OOD detection but also in seen class classification. Besides, we observe that the TBN [36] and Tent [46] methods, although providing satisfactory performance on the corrupted dataset, could harm the segmentation performance on the original dataset. In contrast, the performance gain of our method remains relatively stable across datasets with different levels of domain shift.

Table 6: Segmentation performance evaluation on seen classes in the original FS Static dataset and its corrupted version. Comparison is made with a previous OOD detection method [44] and two test-time adaptation methods [36, 46] using mIoU(%) and mAcc(%) as metrics.

| | mIoU ↑ | | mAcc ↑ | |
|---|---|---|---|---|
| | FS Static | FS Static + C | FS Static | FS Static + C |
| PEBAL [44] | 87.52 | 58.83 | 97.56 | 72.32 |
| + ATTA (Ours) | 87.44 | **85.64** | 97.54 | **97.09** |
| + TBN [36] | 84.88 | 84.02 | 97.13 | 96.70 |
| + Tent [46] | 81.66 | 84.07 | 96.63 | 96.72 |

## C.3 Performance on Isolated Domain Shifts

We conduct additional experiments by introducing smog, color shifting, and Gaussian blur individually to the original FS Static dataset. Results, as shown in Table 7, reveal consistent improvements by our method across isolated domain shifts.

Table 7: We modify the original FS Static dataset by introducing fog, color shifting, and Gaussian blur separately, to analyze model performance on isolated domain shifts. We compare our method with the previous OOD detection method, PEBAL.

|  | Fog | | | Color | | | Blur | | |
|---|---|---|---|---|---|---|---|---|---|
|  | AUC ↑ | AP ↑ | FPR$_{95}$ ↓ | AUC ↑ | AP ↑ | FPR$_{95}$ ↓ | AUC ↑ | AP ↑ | FPR$_{95}$ ↓ |
| PEBAL [44] | 48.37 | 1.58 | 91.82 | 98.58 | 81.93 | 6.26 | 99.46 | 89.46 | 2.07 |
| + ATTA (Ours) | **98.92** | **79.98** | **3.48** | **99.15** | **87.43** | **2.91** | **99.55** | **90.73** | **1.71** |

# D  Additional Results on Dense OOD Detection Benchmarks

## D.1 Results of Other TTA Methods on the Benchmarks

We compare our method with two other TTA methods, TBN [36] and Tent [46], on three OOD detection benchmarks: the Road Anomaly dataset [29], the Fishyscapes Lost & Found dataset [1], and the Fishyscapes Static dataset [1]. Our method consistently improve upon the previous state-of-the-art OOD detection method, PEBAL [44], while other TTA methods tend to degrade the performance of the baseline model, especially in terms of the FPR95 metric. This indicates that traditional TTA methods often indiscriminately reduce the uncertainty or OOD scores of novel classes, resulting in poor OOD detection accuracy. In contrast, our method demonstrates superior performance by effectively handling the domain shift and semantic shift jointly.

Table 8: Comparison of our method with other TTA methods on the OOD detection benchmarks: Road Anomaly dataset, Fishyscapes Lost & Found dataset, and Fishyscapes Static dataset.

| Methods | Road Anomaly | | | FS LostAndFound | | | FS Static | | |
|---|---|---|---|---|---|---|---|---|---|
|  | AUC ↑ | AP ↑ | FPR$_{95}$ ↓ | AUC ↑ | AP ↑ | FPR$_{95}$ ↓ | AUC ↑ | AP ↑ | FPR$_{95}$ ↓ |
| PEBAL [44] | 87.63 | 45.10 | 44.58 | 98.96 | 58.81 | 4.76 | 99.61 | 92.08 | 1.52 |
| + ATTA (Ours) | **92.11** | **59.05** | **33.59** | **99.05** | **65.58** | **4.48** | **99.66** | **93.61** | **1.15** |
| + TBN | 85.56 | 47.81 | 59.43 | 95.40 | 35.01 | 29.37 | 99.05 | 82.45 | 4.05 |
| + Tent | 85.35 | 47.51 | 60.0 | 95.39 | 35.29 | 29.54 | 99.02 | 81.99 | 4.17 |

## D.2 Results with ResNet 101 backbone

To assess the generalization capability of our method, we conduct experiments using a different segmentation backbone: DeepLabv3+ with ResNet101. Since some previous work, such as [44], does not release their pretrained models on this backbone, we compare our method with two simpler OOD detection baselines, namely Max Logit [15] and Energy Score [30], which do not require additional training. The results in Table 9 demonstrate that our method consistently improves upon these baseline methods across three evaluation metrics.

Table 9: Evaluation of our method with the ResNet101 backbone on the Road Anomaly dataset. Our method consistently improves upon the Max Logit and Energy Score baselines.

|  | AUC ↑ | AP ↑ | FPR$_{95}$ ↓ |
|---|---|---|---|
| Max Logit [15] | 76.59 | 22.97 | 66.40 |
| + ATTA (Ours) | **84.39** | **37.84** | **49.72** |
| Energy [30] | 77.24 | 24.05 | 66.20 |
| + ATTA (Ours) | **84.02** | **38.95** | **51.27** |

## D.3 Results on the SMIYC benchmark

We evaluate our method on the SegmentMeIfYouCan (SMIYC) [3] benchmark, which particularly includes significant domain shifts such as variations in illumination and weather. We submit our outputs to the benchmark test set and present the results in Table 10. Our experiments on the RoadAnomaly21 and RoadObstacle21 datasets, both part of the SMIYC benchmark, demonstrate significant improvements over the previous SOTA method PEBAL [44]. Notably, in the RoadObstacle21 dataset, PEBAL's performance is hampered by a lack of robustness, whereas our method has increased the AUPRC score from 5.0% to 76.5%. This validates our method's adaptability to substantial domain shifts.

Table 10: Results on the SMIYC official test benchmark. The results for our model were obtained by submitting the model outputs to the benchmark organizer, as required. The results for PEBAL were taken from the benchmark's official website.

| RoadAnomaly21 | AP↑ | FPR$_{95}$ ↓ | sIoU↑ | PPV↑ | F1↑ |
|---|---|---|---|---|---|
| PEBAL [44] | 49.1 | 40.8 | 38.9 | 27.2 | 14.5 |
| + ATTA (Ours) | **67.0** | **31.6** | **44.6** | **29.6** | **20.6** |
| RoadObstacle21 | AP↑ | FPR$_{95}$ ↓ | sIoU↑ | PPV↑ | F1↑ |
| PEBAL [44] | 5.0 | 12.7 | 29.9 | 7.6 | 5.5 |
| + ATTA (Ours) | **76.5** | **2.8** | **43.9** | **37.7** | **36.6** |

## D.4 Results on the Fishyscapes Online Testing Set

We evaluate our method on the Fishyscapes Online Testing set [1]. As presented in Table 11, our method outperforms the previous state-of-the-art, PEBAL [44].

Table 11: Results on the Fishyscapes online test benchmark. The results for our model were obtained by submitting it to the benchmark organizer. The results for PEBAL were taken from their published paper.

| Online FS Lost & Found | AP↑ | FPR$_{95}$ ↓ |
|---|---|---|
| PEBAL [44] | 44.17 | 7.58 |
| + ATTA (Ours) | **55.94** | **4.66** |
| Online FS Static | AP↑ | FPR$_{95}$ ↓ |
| PEBAL [44] | 92.38 | 1.73 |
| + ATTA (Ours) | **94.68** | **0.68** |

## E  Additional Ablation Results

In this section, we first evaluate our method's inference overhead, then analyze the efficacy of our model components including trainable parameters, optimization techniques, and loss weight.

**Analysis of Inference Time**   We evaluate the average inference time for each image. As shown in Table 12, our method is only 2.25 times slower than direct inference, and faster than another test-time adaptation method, Tent [46], and some ood detection methods with posthoc operations: ODIN [27], Mahalanobis Distance [25] and Synboost [10]. This efficiency is attributed to our design, which

Table 12: Comparison of inference time (seconds per image). We calculate the complete time from input to the final prediction and/or ood score. Experiments are conducted on one NVIDIA TITAN Xp device, and results are averaged over all images in the FS Lost & Found validation set, with image size (1024 x 2048).

| Methods | Direct Inference | ATTA (Ours) | ATTA (Ours) w/o SBN | Tent | ODIN | SynBoost | Mahalanobis |
|---|---|---|---|---|---|---|---|
| Time (s) | 1.2 | 2.7 | 1.5 | 5.1 | 9.2 | 3.0 | 224.2 |

updates only once per image and confines learnable parameters to the classifier block. The latter design enables us to perform backward and the subsequent forward pass only on the classifier block, and is the main reason we achieve much faster inference than Tent. Furthermore, in scenarios where data from the same domain are known, we can reduce the computation by performing domain-shift detection only once, maintaining the variable for subsequent images. To illustrate this, we show the inference time without domain-shift detection, which further reduces it to 1.25 times the direct inference speed. Practically, our episodic training model allows for parallel inference across multiple processors. With ongoing hardware advancements, we anticipate a further reduction in the time gap.

**Analysis of Memory Overhead**  For our method, the memory consumption does increase temporarily for storing activation memory during loss computation but aligns with direct inference after backpropagation. Table 13 below illustrates that our memory requirement is higher than direct inference but significantly more efficient than Tent.

Table 13: Comparison of maximum GPU memory consumption during test time (in MB), using DeepLab v3+ with WideResNet38 on an NVIDIA TITAN Xp, input image size (1024 x 2048).

| Methods | Max Memory (MB) |
|---|---|
| Direct Inference | 1170.2 |
| ATTA (Ours) | 3388.6 |
| Tent | 12796.7 |

**Analysis of Trainable Parameters**  In Table 14, we show the results on the Road Anomaly dataset with different trainable parameters. The configurations include: 'All' (all parameters are trainable), 'Body' (only the feature extraction part of the network is trainable while the final classification layers are fixed), 'Head' (only the final classification layers are trainable), and 'BN affine' (only the parameters of the affine transformation of BN layers are learnable, similar to [46]). We observe that all configurations are plausible in our setting, with the performance difference being less than 1% for most metrics. Notably, by training only the parameters in the head, we enable faster inference for each iteration since we do not need to recalculate the features extracted by the body part.

Table 14: Analysis of the effect of different trainable parameters on the Road Anomaly dataset.

| Trainable parameters | AUC $\uparrow$ | AP $\uparrow$ | FPR$_{95}$ $\downarrow$ |
|---|---|---|---|
| All | 92.20 | 59.60 | 31.99 |
| Body | 92.19 | 59.58 | 32.00 |
| BN affine | 91.73 | 58.68 | 34.30 |
| Head (Ours) | 92.11 | 59.05 | 33.59 |

**Analysis of Episodic Optimization**  As stated in the main text, we adapt the model parameters on each image independently in an episodic manner. To evaluate alternative test-time optimization strategies, we conduct experiments on the Road Anomaly dataset with two variants. In the first variant, we continue updating the model parameters with the incoming test data (denoted as 'Continue'). In the second variant, we processed two images together as a batch (batch size = 2) instead of one image at a time. The results are shown in Table 15. We observed that both variants led to a performance decrease compared to our episodic updating method. This can be attributed to the lower correlation between images in the Road Anomaly dataset. The information learned from previous data potentially negatively affected the model performance on subsequent data, and processing different images together as a batch introduced inconsistencies and inaccuracies in domain-shift identification, impacting the final performance. Consequently, our episodic updating method tends to be more robust, enabling better adaptation and OOD detection in scenarios with less correlated or diverse data.

**Analysis of the outlier class weight**  We also analyze the effect of the outlier class weight in our loss (represented by $\lambda$ in Eq.(10)) by removing this term. The results are shown in Table 16. We note that this weight is designed to address the class imbalance problem between the inlier and outlier classes. In our analysis, we observe that for the Road Anomaly dataset, where the class imbalance is not severe, the performance without the weight is comparable to the performance with the weight.

Table 15: Analysis of optimization manner on the Road Anomaly dataset. For the "Continue" updating, the data order may affect the output. Hence, we examine with two different data orders.

| Training manner | Batch Size | AUC ↑ | AP ↑ | FPR$_{95}$ ↓ |
|---|---|---|---|---|
| Continue - order 1 | 1 | 87.19 | 57.3 | 67.88 |
| Continue - order 2 | 1 | 89.09 | 58.14 | 50.77 |
| Episodic | 2 | 89.57 | 50.66 | 39.99 |
| Episodic | 1 | **92.11** | **59.05** | **33.59** |

However, for the Fishyscapes Lost & Found and Fishyscapes Static datasets, where the proportion of outlier objects in the images is significantly lower, there is a notable performance gap. Therefore, our design of outlier class weight enables a more robust performance across different datasets with varying proportions of outlier objects.

Table 16: Analysis of the class weight on three datasets with varying proportions of outlier objects.

| Methods | Road Anomaly | | | FS LostAndFound | | | FS Static | | |
|---|---|---|---|---|---|---|---|---|---|
| | AUC ↑ | AP ↑ | FPR$_{95}$ ↓ | AUC ↑ | AP ↑ | FPR$_{95}$ ↓ | AUC ↑ | AP ↑ | FPR$_{95}$ ↓ |
| No Weight | 92.05 | 58.39 | 33.47 | 98.78 | 57.60 | 6.28 | 99.57 | 91.62 | 1.76 |
| Weight (Ours) | **92.11** | **59.05** | 33.59 | **99.05** | **65.58** | **4.48** | **99.66** | **93.61** | **1.15** |

## F  Additional Qualitative Results

**Visualization on the FS Static Dataset**  Figure 4 shows additional qualitative results on the FS Static Dataset and its corrupted version. Our method effectively mitigates domain shift effects, as evidenced by clearer separation between inliers (blue) and outliers (orange) in the histograms.

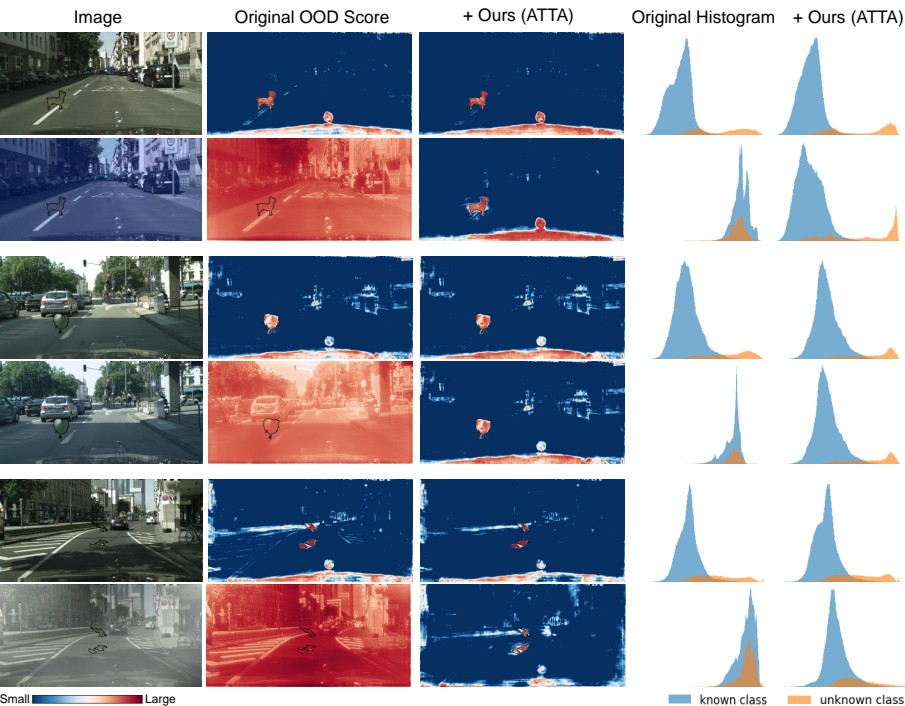

Figure 4: Visualization of the original images in the FS Static dataset and their corrupted versions. We present the OOD score outputs by PEBAL [44] (denoted as 'Original') and our method, along with their corresponding histograms. The ground truth OOD objects are highlighted with black lines.

**Visualization on the Road Anomaly Dataset** As shown in Figure 5, our method effectively reduces the high uncertainty region associated with the inlier classes, and the separation between the inliers (depicted in blue) and outliers (depicted in orange) is more distinct in the histogram.

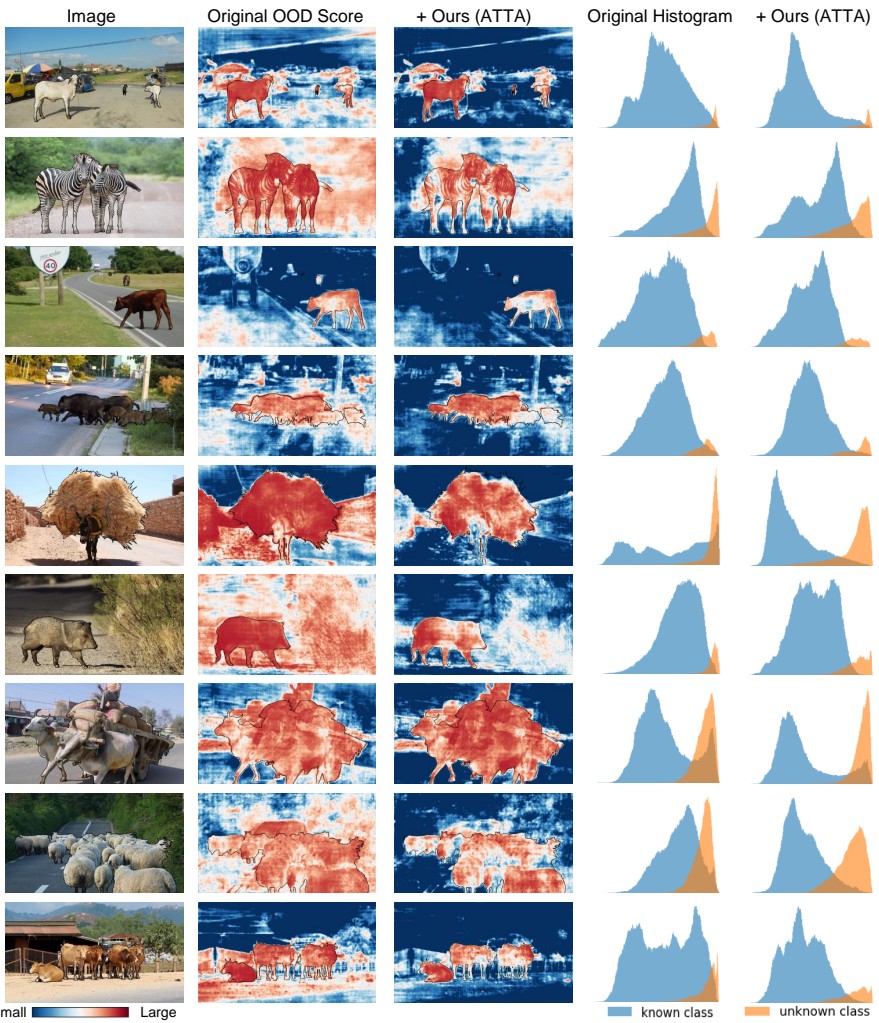

Figure 5: Visualization results on Road Anomaly. Ground truth OOD objects are marked in black.

**Visualization Results on SMIYC validation sets.** Figure 6 presents qualitative results on the SMIYC validation set. Our method clearly reduces the high uncertainty region of inlier classes.

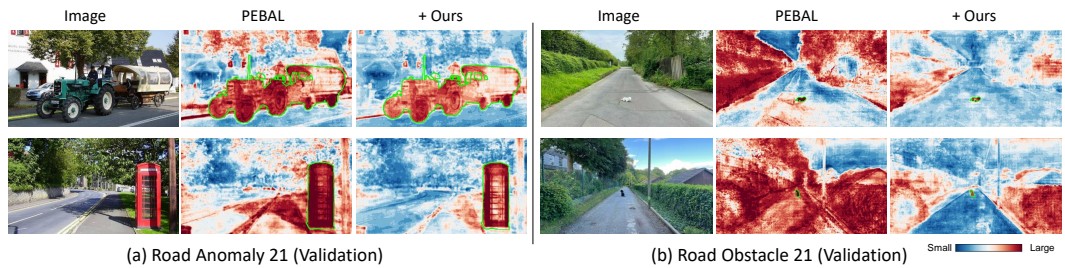

Figure 6: Visualization results on the SMIYC validation set, including (a) the Road Anomaly 21 dataset, and (b) the Road Obstacle 21 dataset. Ground truth OOD objects are marked in green.

# G   Discussion

## G.1   Specific Domain Shift in the Road Anomaly Dataset

The specific 'domain shift' in the Road Anomaly dataset, as compared to the 'Cityscapes' dataset, encompasses adverse road conditions, diverse weather and lighting conditions, and various camera perspectives and conditions. Here, we detail each of these aspects:

**Adverse Road Conditions**   The Cityscapes dataset mainly focuses on urban scenes with well-paved roads and uniform coloration, whereas the Road Anomaly dataset extends to include more diverse locations like villages and mountains. These rural pathways exhibit varied textures and colors due to different materials, wear, soil types, and vegetation. Examples in Fig. 3 illustrate this contrast, with the visualization of Fishyscapes Lost & Found / Static serving as a reference for typical Cityscapes road conditions. Besides, we see that previous OOD detection methods often mistake these variations for anomalies, while our algorithm reduces such errors, showing better adaptation to the domain shift in road conditions. We refer to Appendix Fig. 5 for more visualization cases on the Road Anomaly dataset, where other examples with various road conditions can be seen.

**Weather and Lighting Variations**   The Road Anomaly dataset encompasses diverse weather conditions not presented in Cityscapes, including snowy, rainy, foggy weather, and nighttime scenes. These variations affect not only the road but also the surrounding areas and the sky, leading to more false positive errors in existing OOD detection models. Examples of these weather-related differences and their effects on OOD detection can be found in Appendix Fig. 7. Our method strives to account for these variations, enhancing the model's adaptability to changes in weather and lighting.

**Various Camera Conditions**   Besides differences in content, the Cityscapes and Road Anomaly datasets also diverge in the conditions under which the images were captured. In the Road Anomaly dataset, images may be captured from various angles and locations, such as alongside the road, which differs from the typical road-centered perspective in Cityscapes. This variance can disrupt previously learned biases in road surface predictions. Additionally, some Road Anomaly images demonstrate a focus effect where the background is intentionally blurred, an effect not commonly seen in Cityscapes. This can lead to false positive errors by initial OOD detection models, as shown in Figure 3, illustrating the sensitivity of models to different camera conditions.

In summary, the Road Anomaly dataset's construction, with images gathered from various internet sources, reflects a real-world scenario that involves complex domain-level distribution shifts. These shifts present an intriguing problem for existing OOD detection methods. Our research contributes to understanding and addressing these domain shifts within the dataset.

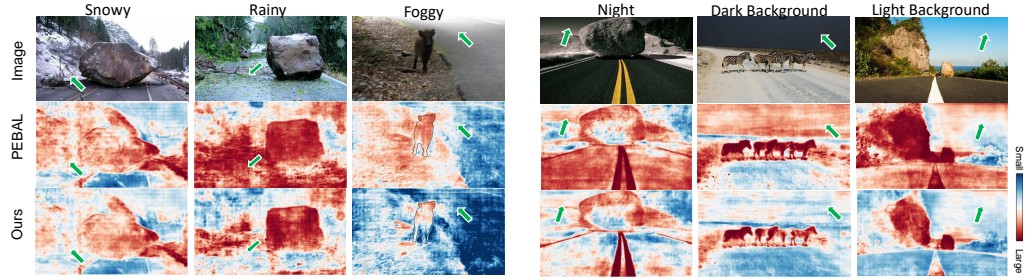

Figure 7: First Row: Examples of images from the Road Anomaly Dataset, showcasing various weather conditions including snowy, rainy, and smoggy scenes, along with differing lighting conditions. Second Row: Corresponding OOD score maps generated by the prior OOD detection method (PEBAL). Third Row: OOD scores after applying our ATTA algorithm for model adaptation. Green arrows indicate regions with prominent domain-shift, highlighting that the previous method in the second row assigned high scores to these regions.

## G.2 Architectures without BN

While we primarily target models with BN, it's worth noting that most modern networks utilize certain types of normalization techniques including Instance Normalization (IN) and Layer Normalization (LN). For the case of IN and LN, we note that they inherently offer more stability across domain shifts, thus saving special adjustments in our first cascaded stage. This is because, unlike BN, they normalize individual samples, making them less sensitive to variations between domains. We have also empirically tested our anomaly-aware self-training on the mentioned Segmenter backbone [41] which employs layer normalization. Results shown in Table 17 demonstrate the enhanced performance when incorporating our method.

Table 17: Results on the RoadAnomaly Dataset with Segmenter. We use the pre-trained model weights on the Cityscapes dataset provided by [41].

|  | AUC $\uparrow$ | AP $\uparrow$ | FPR$_{95}$ $\downarrow$ |
|---|---|---|---|
| Energy | 95.43 | 75.60 | 19.76 |
| +ATTA (Ours) | **96.49** | **84.77** | **18.56** |
| Max logit | 94.81 | 71.65 | 20.96 |
| +ATTA (Ours) | **96.08** | **81.88** | **19.19** |

Still, we note that, since our goal is to adapt at the test phase without retraining, and that BN is prevalent in most network architectures, it's practical to recognize BN's ubiquity and design accordingly, especially in our task, where many methods [44, 4, 1] are based on the Deep Lab V3+ structure, which utilizes BN.

In summary, our framework is broadly applicable across the vast majority of modern neural network architectures that employ some form of normalization. The test-time adaptation of models without normalization layers remains an open UDA problem, which is out of the scope of this work.

