# OpenReview forum: "ATTA: Anomaly-aware Test-Time Adaptation for Out-of-Distribution Detection in Segmentation"
_NeurIPS.cc/2023/Conference — NeurIPS 2023 poster_

### Official Review · Reviewer_XNCt · 2023-06-29

**Soundness:** 2 fair
**Presentation:** 2 fair
**Contribution:** 3 good
**Rating:** 5
**Confidence:** 4

**Summary:**

The manuscript considers dense OOD detection under domain shift. The manuscript shows that the contemporary methods for dense OOD detection experience performance drop under domain shift and propose an adaptation framework to mitigate the issue. The proposed framework has two steps. The first step determines whether domain shift exists and attempts to reduce it by adapting the statistics of batch normalization layers. The second step iteratively adapts network parameters by optimizing outlier-aware self-supervised loss. Outlier identification during the test-time training is done by renormalizing arbitrary OOD scores. The resulting method can accommodate various anomaly detectors and achieves competitive results in considered benchmarks for dense OOD detection with and without domain shift.

**Strengths:**

S1. The road driving scenes indeed experience domain shift. E.g. changes in weather conditions and geolocation affect the captured scenes. Hence, the task of dense OOD detection under domain shift makes sense and appears to be novel.

S2. The developed method can accommodate multiple methods for dense OOD detection, which advocates for general applicability.

S3. The method achieves competitive results in OOD detection in road driving scenarios under considered domain shifts.

**Weaknesses:**

W1. Contemporary works for dense predictions in traffic scenes consider four main types of domain shift: geolocation, weather conditions [a], day-to-night [b], and synthetic to real [c]. Yet, the manuscript deals only with dense OOD detection under different geolocation (RoadAnomaly). Moreover, FS Static-C contains blurred and colour-jittered images which may not adequately reflect real-world domain shifts in traffic scenes. Authors should experimentally cover all possible domain shifts in traffic scenes.

W2. The proposed framework requires gradient-based optimisation (and hence backpropagation) during the inference. The manuscript should report the time and memory overhead as in [14,21]. Adding considerable computational burden may make the proposed method inapplicable to the considered application.

W3. The proposed framework may only work with models which have batchnorm layers (Sec. 3.3). The manuscript should reflect on models which do not use batch norm layers (eg. attention-based architectures [d]).

W4. The manuscript misses relevant works [e,f,g].

[a] Christos Sakaridis, Dengxin Dai, Luc Van Gool: Semantic Foggy Scene Understanding with Synthetic Data. Int. J. Comput. Vis. (2018)

[b] Christos Sakaridis, Dengxin Dai, Luc Van Gool:
Guided Curriculum Model Adaptation and Uncertainty-AwareEvaluation for Semantic Nighttime Image Segmentation. ICCV 2019.

[c] Lukas Hoyer, Dengxin Dai, Luc Van Gool: DAFormer: Improving Network Architectures and Training Strategies for Domain-Adaptive Semantic Segmentation. CVPR 2022.

[d] Robin Strudel, Ricardo Garcia Pinel, Ivan Laptev, Cordelia Schmid: Segmenter: Transformer for Semantic Segmentation. ICCV 2021.

[e] Shu Kong, Deva Ramanan: OpenGAN: Open-Set Recognition via Open Data Generation. ICCV 2021

[f] Matej Grcic, Petra Bevandic, Sinisa Segvic: Dense Open-set Recognition with Synthetic Outliers Generated by Real NVP. VISAPP 2021.

[g] Chen Liang, Wenguan Wang, Jiaxu Miao, Yi Yang: GMMSeg: Gaussian Mixture based Generative Semantic Segmentation Models. NeurIPS 2022

**Questions:**

C1. More detailed Fig. 2 may improve clarity.

C2. Since the manuscript introduces a new task it is very important to establish proper evaluation experiments. How does the model perform under different weather and illumination?  E.g. when applying transformations similar to [a] which imitate rain/fog.

C3. The SMIYC benchmark [3] contains images with different illumination (night) and weather (snow, fog). How does the method perform on this benchmark?

**Limitations:**

The manuscript did not reflect on the limitations of the model. One possible limitation may be described in W3.

---

> ### Author Rebuttal · Authors · 2023-08-10
>
> Thank you for recognizing the novelty and applicability of our work. We have added results on the SMIYC benchmark, inference time analysis, and a refined Figure 2 in our general response and attached PDF. We appreciate the reviewer for pointing out missing relevant works and will include them in the related work section. We address your other concerns below.
>
> ## W1: Different types of domain shift
>
> In this work, we focus on the dense OOD detection problem and consider realistic domain shifts likely to exist in this task. We agree that geolocation, weather conditions, and day-to-night shifts are relevant. However, the synthetic-to-real shift is less applicable in our context. Besides, we emphasize that the adopted datasets, namely RoadAnomaly and FS Static-C datasets, do cover various domain adaptations. Specifically,
>
> - For the RoadAnomaly dataset, images vary significantly from specialized collections in Cityscapes. Differences include geographical location, weather, lighting (day/night), road conditions, camera settings, etc. Please refer to our reply for Reviewer 7M1E - 2 for the detailed analysis and the submitted pdf for image examples.
>
> - FS Static-C is a synthetic dataset comprised of random smog, color shifting, and Gaussian blur. The random smog simulates weather conditions, while the color shifting may represent various lighting scenarios such as dusk or sunset, and the Gaussian blur could signify different camera settings. Although these transformations might not perfectly align with real-world scenarios, they are commonly used in the domain adaptation literature [7,27,45] and reflect the robustness of different models against domain shifts, which is the primal focus of this dataset in our paper.
>
> We also performed additional experiments on the SMIYC benchmarks, containing variations in geolocation, weather condition, and night information. Our results significantly improved OOD detection, further validating the effectiveness of our method (See General Response for details).
>
> We note that our work is different from those specifically targeting domain adaptation, and we focus more on the pronounced issues within the OOD detection task, such as ensuring the model works both with and without domain shift, without affecting the prediction of OOD scores.
>
> ## W2: Memory Overhead
>
> Memory consumption does increase temporarily for storing activation memory during loss computation but aligns with direct inference after backpropagation. Table E1 below illustrates that our memory requirement is higher than direct inference but significantly more efficient than Tent.
>
> Table E1: Comparison of maximum GPU memory consumption during test time (in MB), using DeepLab v3+ with WideResNet34 on an NVIDIA TITAN Xp, input image size (1024 x 2048).
> |                  | Max Memory (MB) |
> |------------------|-----------------|
> | Direct Inference | 1170.2          |
> | ATTA (Ours)      | 3388.6          |
> | Tent             | 12796.7         |
>
> ## W3: Architectures without BN
>
> While we primarily target models with BN, it's worth noting that most modern networks utilize certain types of normalization techniques including Instance Normalization (IN) and Layer Normalization (LN). For the case of IN and LN, we note that they inherently offer more stability across domain shifts, thus saving special adjustments in our first cascaded stage.  This is because, unlike BN, they normalize individual samples, making them less sensitive to variations between domains [e1].  We have also empirically tested our anomaly-aware self-training on the mentioned Segmenter backbone [d] which employs layer normalization. Results shown in Table E2 demonstrate the enhanced performance when incorporating our method.
>
> Still, we note that, since our goal is to adapt at the test phase without retraining, and that BN is prevalent in most network architectures, it's practical to recognize BN's ubiquity and design accordingly, especially in our task, where many methods [5,21,42] are based on the Deep Lab V3+ structure, which utilizes BN.
>
> In summary, our framework is broadly applicable across the vast majority of modern neural network architectures that employ some form of normalization.  The test-time adaptation of models without normalization layers remains an open UDA problem, which is out of the scope of this work.
>
> Table E2. Results on the RoadAnomaly Dataset with Segmenter. We use the pre-trained model weights on the Cityscapes dataset provided by [d].
>
> |              | AUC $\uparrow$      | AP $\uparrow$       | FPR $\downarrow$     |
> |--------------|-----------|-----------|-----------|
> | Energy       | 95.43     | 75.60     | 19.76     |
> | +ATTA (Ours) | **96.49** | **84.77** | **18.56** |
> | Max logit    | 94.81     | 71.65     | 20.96     |
> | +ATTA (Ours) | **96.08** | **81.88** | **19.19** |
>
> ## C2: Different weather and illumination
> We have considered various domain shifts in our dataset, including different weather and illumination conditions (cf. the response to W1), which establishes a proper evaluation protocol. In particular,  we have actively considered foggy conditions (cf. Appendix B) and conducted extensive testing of our model under various weather and illumination scenarios (cf. General Response 1).
>
> We appreciate the reference to Sakaridis et al. [a]. However, it's not directly applicable to our task, as the foggy Cityscapes dataset lacks the novel class information needed for OOD detection evaluation. Additionally, the provided code to generate fog from [a] seems to require specific disparity and camera information, which may not be accessible in our OOD test dataset.
>
> ## Limitation
> We'd like to kindly note that we discuss some limitations of our method in the conclusion. We have provided a response to your concern in W3, demonstrating that our model can be applied to network architectures without BN.
>
> [e1] Shuaicheng Niu et al. Towards Stable Test-time Adaptation in Dynamic Wild World. ICLR, 2023.

---

> ### Comment · Reviewer_XNCt · 2023-08-17
> **Post rebuttal**
>
> The presented response addresses most of my concerns (W2, W3, W4.) and dense OOD detection under domain shift is an attractive research topic.
>
> However, the current state of the manuscript does not consider domain shifts in isolation which is common in the field of domain adaptation e.g. [1]. On the contrary, RoadAnomaly mixes images with different domain shifts. The dataset should be sorted according to the domain (and even enlarged) to show more informative results.
>
> [1] Christos Sakaridis, Dengxin Dai, Luc Van Gool: ACDC: The Adverse Conditions Dataset with Correspondences for Semantic Driving Scene Understanding. ICCV 2021.
>
> I will increase my score to 4, but I still believe the manuscript should undergo a major revision and another round of reviews.

---

> > ### Author Response · Authors · 2023-08-20
> > **Response to Reviewer XNCt**
> >
> > Thank you for your comment and for recognizing the appeal of dense OOD detection under domain shift.
> >
> > Regarding your concern, we would like to clarify that our primary focus is on enhancing OOD detection across various levels of domain shifts, rather than adhering strictly to traditional domain adaptation. This approach, which includes consideration of datasets with and without significant domain shifts in isolation, emphasizes our model's practical ability to cope with different degrees of shifts. We believe this perspective is more aligned with the essential and real-world concerns of OOD detection.
> >
> > Concerning the isolation of different types of domain shifts, we appreciate the reference to datasets like [1] that categorize shifts. Yet, in current dense OOD detection datasets, isolating specific domain shifts is a practical challenge. This complexity stems from the lack of explicit consideration of domain shifts during dataset construction. For instance, in datasets like Road Anomaly, various shifts may be intermingled within an image—such as different road conditions with diverse weather—making clear division difficult.
> >
> > To provide information on our model's behavior under various domain-shift types, we've included visualizations in the manuscript (Fig. 2) and the Attached PDF (Fig. 2), demonstrating our method's stable performance across different scenarios. In response to your suggestion, we also conduct additional experiments by introducing smog, color shifting, and Gaussian blur individually to the original FS Static dataset. Results, as shown in Table E3, reveal consistent improvements by our method across isolated domain shifts.
> >
> > We share the hope for future datasets with clearly defined domain shifts and novel classes to further study this evolving field of OOD detection. However, such exploration is beyond the scope of this paper, and we look forward to benchmarks that will facilitate research in this vital area.
> >
> > Table E3. We modify the original FS Static dataset by introducing fog, color shifting, and Gaussian blur separately, to analyze model performance on isolated domain shifts. We compare our method with the previous OOD detection method, PEBAL.
> >
> > |              | Fog   |       |       | Color| |       | Blur |       |       |
> > |--------------|-------|-------|-------|----------------|-------|-------|---------------|-------|-------|
> > |              |  AUC $\uparrow$ |   AP $\uparrow$ | FPR95 $\downarrow$ |       AUC  $\uparrow$    |   AP $\uparrow$ | FPR95$\downarrow$ |      AUC  $\uparrow$    |   AP $\uparrow$ | FPR95$\downarrow$ |
> > | PEBAL        | 48.37 |  1.58 | 91.82 |      98.58     | 81.93 |  6.26 |     99.46     | 89.46 |  2.07 |
> > | PEBAL + Ours | 98.92 | 79.98 |  3.48 |      99.15     | 87.43 |  2.91 |     99.55     | 90.73 |  1.71 |

---

> > > ### Comment · Reviewer_XNCt · 2023-08-20
> > > **On domain shifts**
> > >
> > > The additional response, in particular Table E3 alleviated my concerns. I hope the analysis of domain shifts in isolation is included in the final version of the manuscript. I increased the score to 5.

---

> > > > ### Author Response · Authors · 2023-08-21
> > > > **Response to Reviewer XNCt**
> > > >
> > > > Thank you for your response and for increasing the score. We appreciate your insights and will include the analysis of domain shifts in isolation in the revised manuscript.

---

### Official Review · Reviewer_C29A · 2023-07-05

**Soundness:** 2 fair
**Presentation:** 3 good
**Contribution:** 2 fair
**Rating:** 5
**Confidence:** 3

**Summary:**

This paper proposes ATTA (Anomaly-aware Test-Time Adaptation), which introduces test-time domain adaptation (TTA) for anomaly segmentation. As a result, anomaly segmentation can be performed well even in a harsher environment where domain shift and semantic shift occur simultaneously. To create an environment with domain shift, the authors create the FS Static -C (Corrupted FS Static) dataset, and develope a Selective Test-Time Batch Normalization (SBN) method to propose a new TTA. They also introduce self-supervised learning using GMM.

**Strengths:**

1. The authors show existing methods are vulnerable to domain shift by creating the FS Static -C dataset and experimenting with it.
2. The effectiveness of TTA in the field of anomaly segmentation is demonstrated experimentally (Table 1).
3. ATTA with SBN and self-supervised learning via GMM, shows better performance than TBN or Tent, which were used in the existing vision field.
4. The authors provide a mathematically convincing motivation.


**Weaknesses:**

1.	Table 1 shows that ATTA is exposed to the FS Static -C dataset at test time, and the FPR is re-measured for FS Static -C and the original dataset (FS Static). Therefore, the OOD that must be classified should be seen by the model, so the result should be good. As FS Static -C is a variation of FS Static, it can have a significant impact on the performance of the original dataset. In order to make a meaningful comparison, at least other methods such as Meta-OoD should also be exposed to the FS Static -C dataset and then compared.
2.	The contribution and effect of ATTA is unclear. Using OOD data to improve detection performance has been proposed in the past (e.g., Outlier Exposure[1], Meta-OoD[2], etc.). ATTA also eventually exposes the model to additional OOD data to improve detection performance. However, it is necessary to add a description of the advantages that ATTA has over existing methods by exposing it to the test time. For example, it can be shown in a graph that the segmentation performance increases as the batch progresses during the test time. It is also necessary to show how much the performance is improved through TTA on other datasets other than FS Static -C.
3.	It is unclear whether a fair comparison was made with the OOD data using methods (e.g., Meta-OoD, Synboost, DenseHybrid) in Table 2. Since TTA is able to obtain more information than MSP, Entropy, Mahalanobis and post hocs, ATTA should be superior to these methods that do not use additional OOD data. Therefore, the main competitors of ATTA are methods that utilize additional OOD data in learning (e.g., Meta-OoD, Synboost, DenseHybrid). The authors' method ATTA uses the FS Static -C dataset as an OOD data for training, but the OOD data used for training in the existing methods mentioned above is not specified. Therefore, it is necessary to perform an ablation study to determine whether the superior performance of ATTA is the effect of ATTA, or simply the effect of data augmentation caused by the introduction of the FS Static -C dataset. In addition, ATTA adopted PEBAL as a partner and compared the performance of PEBAL, which was trained on COCO as an additional OOD, with the existing report. PEBAL should also be compared after being exposed to the FS Static -C dataset (excluding ATTA).
4.	The metrics used are AP, AUROC, and FPR95. However, except for AP (Average Precision), these metrics are mainly used in OOD classification rather than anomaly segmentation. It is also necessary to compare the sIoU, PPV, and F1 metrics proposed in the benchmark paper [3] mentioned by the authors in related works.
5.	As the authors mentioned in the Conclusion, TTA is used, so the learning is performed simultaneously at the test time, so the inference time is slow. It is necessary to check if the overhead is too large by comparing the inference time for each method. In particular, GMM clustering must be performed for each image, and it is necessary to check if the overhead due to this is excessive.

[1] Hendrycks, Dan, Mantas Mazeika, and Thomas Dietterich. "Deep anomaly detection with outlier exposure." arXiv preprint arXiv:1812.04606 (2018).

[2] Chan, Robin, Matthias Rottmann, and Hanno Gottschalk. "Entropy maximization and meta classification for out-of-distribution detection in semantic segmentation." Proceedings of the ieee/cvf international conference on computer vision. 2021.

[3] Chan, Robin, et al. "Segmentmeifyoucan: A benchmark for anomaly segmentation." arXiv preprint arXiv:2104.14812 (2021).


**Questions:**

1. Is the FS Static -C dataset is just a data augmentation of the original FS Static dataset? Or is there a significant difference enough to be called another 'dataset'?
2. Is there a performance change when ATTA is trained on the original FS Static dataset rather than the FS Static -C dataset?
3. Is there a performance change when K-means is used instead of GMM clustering to separate inlier and outlier clusters from the OOD score set G for self-supervised learning?


**Limitations:**

1. The authors have well summarized the common features of the OOD field that are affected by the performance of the backbone, and the weakness that the inference is slow because learning is performed during the test time when TTA is used.
2. The inference time is expected to be slow, so it is necessary to compare the inference time for each method to check if the overhead is too large.

---

> ### Author Rebuttal · Authors · 2023-08-10
>
> We thank the reviewer for the time and detailed feedback. Unfortunately, there appears to be some misunderstanding regarding our method, which we will clarify in the following responses.
>
> ## W1: Clarification of the FS Static -C dataset
>
> We would like to clarify that our comparison is indeed fair with other OoD methods, as our approach does not expose the model to any additional information than previous methods. Specifically, our algorithm only encounters online unlabeled image data during inference, a procedure that is also followed by other OOD methods, including Meta-OoD. FS Static-C is merely a testing dataset used to examine the performance of existing OOD methods when a domain shift is introduced. Our algorithm does not utilize any additional information beyond the testing images. Any stated impact of FS Static-C on FS Static appears to be a misunderstanding.
>
> ## W2: Our Contribution compared to existing OOD methods
>
> **Our Contribution:** Our ATTA method leverages online unlabeled test images to enhance OOD detection capacity in potential domain-shift scenarios. This stands apart from previous work in two key aspects:
>
> - *We Do Not Rely on Additional Data:* We exclusively utilize online unlabeled test image, which is naturally available during inference, reflecting the characteristics of the test domain. This presents a unique advantage to our method, as methods like Outlier Exposure and Meta-OoD depend on utilizing labeled OOD data during training and are not applicable to our online and unlabeled scenarios.
>
> - *We Explicitly Address Domain-Shift:* This is an area often overlooked by existing OOD detection methods. Our experiments (see Table 1) underline the fragility of prior methods when faced with domain shifts and validate the effectiveness of our approach in these situations.
>
> **Segmentation performance as batch progresses:** We appreciate the suggestion to showcase how segmentation performance might evolve with more data. However, it's worth noting that our design handles different batches of images individually (cf. Sec 3.2), reflecting the reality that different domains may be represented in each batch.
>
> **Performance of TTA on other datasets:**  In addition to FS Static-C, we have evaluated both our method and other test time adaptation techniques (e.g., TBN, Tent) across several datasets, including the Road Anomaly dataset, Fishyscapes Static dataset, and Fishyscapes Lost And Found dataset (results detailed in Table 2 and Appendix Table 2). We have further included extra results on the SegmentMeIfYouCan (SMIYC) benchmark, outlined in the General Response. For all these datasets, we see a clear performance increase by combining our algorithm, which indicates its efficacy.
>
> ## W3: Fair Comparison
>
> **Fair Comparison:** As detailed in our previous response to W1, our method doesn't utilize FS Static-C or any additional OOD data for training. Instead, we only employ the online unlabeled test image that is naturally available during inference. In Table 2, we've maintained a fair comparison by evaluating our approach alongside various OOD methods. This includes those that use additional OOD data during training (e.g., PEBAL), and those that do not (e.g., Energy score or Max Logit), providing a balanced and fair assessment.
>
> **General Applicability:** We would like to note that our method is not designed specifically for PEBAL, but is applicable to a wide array of differentiable OOD functions and training strategies (cf. Sec. 3.2). Empirically, we have tested on Max Logit, Energy, PEBAL (as shown in Table 2), as well as Meta-OOD, Entropy, MSP (as detailed in Table B1 in response to reviewer TGMY). These results collectively demonstrate the general applicability of our method.
>
> ## W4: Metric
>
> We acknowledge the reviewer's suggestion. It is worth noting that the metrics we have employed, namely AP, AUROC, and FPR95, are not only standard in anomaly segmentation literature [42, 21, 14, 10] but also aligned with the main metrics of the Fishyscapes benchmarks [1]. Additionally, we have included experiments on the SegmentMeIfYouCan (SMIYC) benchmark, where we assessed the model's performance using the sIoU, PPV, and F1 metrics. Details of these evaluation results can be found in General Response.
>
> ## W5 & Limitation: Time Overhead
>
> In our General Response 2, we demonstrate the inference time of our method, concluding that it is only 1.25-2.25 times the pure inference time. This makes our approach much faster than some other OOD detection methods with post hoc operations. Concerning the GMM clustering overhead, it's worth emphasizing that the impact is minimal in our algorithm. This is primarily due to the fact that the score to be fitted is one-dimensional, and we sample 1% of the total data in clustering. Due to the redundancy in pixel-wise output, this operation does not affect the fitting results. Hence, the overhead due to GMM clustering is not excessive for network inference.
>
> ## Q1: Is FS-Static-C just a data augmentation?
>
> The FS Static-C dataset is utilized to test model performance (as detailed in our previous response), not as 'data augmentation,' which is a technique applied during training. Therefore, the term is not appropriate in this context.
>
> ## Q2: Trainning on original FS Static
>
> As we explained earlier, we do not use any additional data for training, and FS Static -C serves solely as a test set. Therefore, the considered scenario does not apply.
>
> ## Q3: Alternative for GMM clustering
>
> K-means clustering is not suitable for our case, as it assigns each point to the nearest centroid, leading to equally sized clusters. Given that outliers are typically much fewer than inliers, this would result in an inaccurate boundary between inliers and outliers, thereby adversely affecting the performance of the self-training model.

---

> > ### Comment · Reviewer_C29A · 2023-08-17
> >
> > It appears that I initially misunderstood the utilization of the FS Static-C dataset. I commend the authors for providing clarification on their work in the rebuttal. Upon conducting a thorough re-evaluation of both the main manuscript and the authors' rebuttal, I have come to the realization that the authors have diligently tackled the majority of the concerns I had raised. Consequently, I see no justification for maintaining the initial score, and I am inclined to raise it. Nonetheless, I agree that the paper should go through another round of peer review with many changes. Hence, I will refrain from making a significant increase in my score.

---

> > > ### Author Response · Authors · 2023-08-17
> > > **Response to Reviewer C29A**
> > >
> > > Thank you for re-evaluating our paper and for increasing the score.
> > > We would like to clarify that the proposed changes, as detailed in our response to Reviewer TGMY, are specific and minimal. These adjustments are designed to address the comments raised without altering the core of our work, and the results are clearly presented in the rebuttal.

---

### Official Review · Reviewer_7M1E · 2023-07-05

**Soundness:** 3 good
**Presentation:** 3 good
**Contribution:** 3 good
**Rating:** 6
**Confidence:** 3

**Summary:**

This paper focuses on the challenging task of open-set semantic segmentation (i.e., dense out-of-distribution (OOD) detection) with domain shift. It proposes a dual-level test-time adaptation framework to overcome domain shift and semantic shift simultaneously, which leverages low-level feature statistics to detect whether domain shift exists while identifying pixels with semantic shift by utilizing dense high-level feature maps. Specifically, it designs an anomaly-aware self-training component to address potential domain shift and improve its ability to detect novel classes through re-balanced uncertainty minimization. The proposed framework is demonstrated to obtain consistent performance improvements across various baseline models.

**Strengths:**

1. This paper explores the realistic and challenging task of open-set semantic segmentation in a real-world scenario with domain shift, considering the impact of both domain shift and semantic shift (novel class from unknown domain) comprehensively.
2. The method in this paper seems reasonable and the experimental results prove the significant superiority of the proposed framework on several OOD segmentation benchmarks, regardless of with or without domain shifts.
3. The method and math presentation in this paper is generally clear.


**Weaknesses:**

1. The visualization of the method in Fig.2 appears overly simplistic and fails to highlight the key components of the proposed framework effectively.
2. How to choose the parameters of the GMM in L220? Is the performance sensitive to their variations? The hyperparameter experiment of these parameters would be desirable.
3. Fig.1 intuitively demonstrates a "domain shift" (fog), but through the visual experiment in Fig.3, I can not understand what the specific "domain shift" that exists in the Road Anomaly dataset for the semantic segmentation is. This casts doubt on the practical application of this paper, despite the experimental evidence confirming the effectiveness of the proposed framework.
4. All the formulas in this paper have not been correctly punctuated.


**Questions:**

Please refer to the weakness part.

**Limitations:**

Do not apply.

---

> ### Author Rebuttal · Authors · 2023-08-09
>
> Thank you for your constructive feedback and recognition of our work. To address your concerns, we have revised Figure 2 to better highlight the key components of our proposed framework and the overview of our methodology. The updated visualization can be found in the attached PDF. Additionally, we acknowledge the issue with punctuation in the formulas and will ensure that it is corrected in the revised version of the paper. We respond to your other concerns in the following.
>
> ## 1. Parameters of GMM:
>
> >How to choose the parameters of the GMM in L220? Is the performance sensitive to their variations? The hyperparameter experiment of these parameters would be desirable.
>
> We apologize for any confusion. The parameters of our two-component GMM, specifically $\pi_1, \pi_2, \mu_1,\mu_2,\sigma_1,\sigma_2$ in Line 220, are not determined by manual selection, but rather estimated by fitting the OOD scores to the GMM (its two-component structure is fixed), as described in line 214. This fitting process is conducted using the Expectation-Maximization algorithm, facilitated by the scikit-learn package [1]. The means for the GMM components are initialized in a manner that avoids trivial issues with multiple peaks in the inlier distribution (detailed in Appendix B), and the algorithm's convergence parameters are kept at their default settings within the library.
>
> [1] https://scikit-learn.org/stable/modules/generated/sklearn.mixture.GaussianMixture.html
>
> ## 2. Specific "Domain Shift" in the Road Anomaly Dataset
> >Fig.1 intuitively demonstrates a "domain shift" (fog), but through the visual experiment in Fig.3, I can not understand what the specific "domain shift" that exists in the Road Anomaly dataset for the semantic segmentation is. This casts doubt on the practical application of this paper, despite the experimental evidence confirming the effectiveness of the proposed framework.
>
> The specific 'domain shift' in the Road Anomaly dataset, as compared to the 'Cityscapes' dataset, encompasses adverse road conditions, diverse weather and lighting conditions, and various camera perspectives and conditions.  Here, we detail each of these aspects:
>
> - **Adverse Road Conditions**: The Cityscapes dataset mainly focuses on urban scenes with well-paved roads and uniform coloration, whereas the Road Anomaly dataset extends to include more diverse locations like villages and mountains. These rural pathways exhibit varied textures and colors due to different materials, wear, soil types, and vegetation. Examples in Fig. 3 illustrate this contrast, with the Visualization of Fishyscapes Lost & Found / Static serving as a reference for typical Cityscapes road conditions. Besides, we see that previous OOD detection methods often mistake these variations for anomalies, while our algorithm reduces such errors, showing better adaptation to the domain shift in road conditions. We refer to Appendix Fig. 2 for more visualization cases on the Road Anomaly dataset, where other examples with various road conditions can be seen.
>
> - **Weather and Lighting Variations**: The Road Anomaly dataset encompasses diverse weather conditions not presented in Cityscapes, including snowy, rainy, foggy weather, and nighttime scenes. These variations affect not only the road but also the surrounding areas and the sky, leading to more false positive errors in existing OOD detection models. Examples of these weather-related differences and their effects on OOD detection can be found in the attached PDF. Our method strives to account for these variations, enhancing the model's adaptability to changes in weather and lighting.
>
> - **Various Camera Conditions**: Besides differences in content, the Cityscapes and Road Anomaly datasets also diverge in the conditions under which the images were captured. In the Road Anomaly dataset, images may be captured from various angles and locations, such as alongside the road, which differs from the typical road-centered perspective in Cityscapes. This variance can disrupt previously learned biases in road surface predictions. Additionally, some Road Anomaly images demonstrate a focus effect where the background is intentionally blurred, an effect not commonly seen in Cityscapes. This can lead to false positive errors by initial OOD detection models, as shown in Figure 3, illustrating the sensitivity of models to different camera conditions.
>
> In summary, the Road Anomaly dataset's construction, with images gathered from various internet sources, reflects a real-world scenario that involves complex domain-level distribution shifts. These shifts present an intriguing problem for existing OOD detection methods. Our research contributes to understanding and addressing these domain shifts within the dataset. Additional experiments in the SegmentMeIfYouCan benchmark, also characterized by domain-shift variations in road surfaces, lighting, and weather conditions, further substantiate the observations. For a detailed presentation of the results, please refer to General Response 1.

---

> > ### Comment · Reviewer_7M1E · 2023-08-18
> >
> > Thanks for the iilustration on the concept of "Domain Shift". I have read the responses and all my concerns are addressed. I will adjust the rating accordingly

---

### Official Review · Reviewer_TGMY · 2023-07-06

**Soundness:** 3 good
**Presentation:** 2 fair
**Contribution:** 3 good
**Rating:** 5
**Confidence:** 3

**Summary:**

The papers deal with two levels of domain shift in semantic segmentation, namely the domain shift on the semantic pixel level and the domain shift on the image level. The paper argues that the current dense out-of-distribution (OOD) detection methods are particularly vulnerable in the presence of image-level domain shifts and therefore likely to predict wrong OOD scores on the pixel level. Based on this observation, it presents a two-stage test-time adaptation approach. In the first step, it is performed selective test-time batch normalization, forcing more adaptation in the scenarios which are identified to be novel, while n the second step, it is proposed a self-training procedure using an anomaly-aware output map to enhance the dense OOD detection. The approach is evaluated on a realistic domain-shift dataset, as well as, two non-domain-shift datasets where it shows good performance.


While the rebuttal addresses several points of the reviews, the changes and additional experiments discussed are several. The paper should go through another round of peer review with many changes. Therefore, I will not increase my score.

**Strengths:**

- The analysis of how current OOD detection methods perform on datasets with distribution shifts, as shown in Figure 1 and Table 1, is very valuable and shows the importance of the problem studied throughout the paper.

- The idea is easy to follow.

- The combination of two steps, first to reduce the domain gap at the image level and second to improve the dense OOD detection performance of the models, is not well explored in the literature.

- The results are convincing when compared to the prior work.

**Weaknesses:**

- The clarity of the method has place for improvement. For instnace, the notation is one point, e.g. 120 line x^s, x^s not defined, Eq. 1 \mathcal{N} not defined. The method can be understoood in general, it could be further polished to make it easier. There are also minor typos, e.g. line 38, line 245, line 194. Also, KL divergence is not clearly defined in Eq 1.

- In Fig. 1: What is shown in (a) and (b)? A detailed description would be  helpful.

- For comparison with their method, the authors add Tent to PEBAL. An explanation of how this is implemented would be helpful.

- Table 1 compares 6 different OOD detection methods on the synthesised corrupted FS Static (FS Static -C) dataset. It shows a performance drop for all OOD detection methods on the FS Static -C dataset, but applying the proposed approach to PEBAL reduces this drop. It would be interesting to see if a similar effect can be achieved for the remaining 5 OOD detection methods.

**Questions:**

- It is not straightforward to implement the paper. There is no discussion about releasing the code.

**Limitations:**

- Not applicable .

---

> ### Author Rebuttal · Authors · 2023-08-10
>
> We appreciate the reviewer for insightful feedback and recognition of the novel aspects of our work. We are committed to improving the clarity in our final version, and we intend to release the code after the double-blind stage. Below, we respond to the specific concerns raised:
>
> ## W1: Clarity
>
> - *Notation:* We thank the reviewer for the suggestion and will ensure to make the notation more clearly explained in the revised manuscript. Specifically,  in line 120, $x^s$ and $y^s$  denote the input image and corresponding segmentation label in the training data, respectively. In Eq. 1, $\mathcal{N}$ denotes the Normal distribution, and $KL$divergence is defined in its standard form as $KL(P \| Q)=\sum_{x \in \mathcal{X}} P(x) \log \left(\frac{P(x)}{Q(x)}\right)$. This equation serves to calculate the KL divergence between two normal distributions with different parameters, as defined in the manuscript.
>
> - *Typos:* We acknowledge the typos mentioned, specifically in lines 38, 245, and 194, and will ensure they are corrected in the revised manuscript.
>
> ## W2: Detailed Description of Figure 1 Components (a) and (b)
>
> We provide a more detailed description and results analysis of Figure 1 in the following, which will also be incorporated in the revised manuscript's caption.
>
> - In Figure 1 (a), we show the visualization results of the OOD score maps produced by the previous SOTA method (PEBAL) for an original image and its domain-shifted version (corrupted with smog).  The first column displays both images, while the second and third columns depict the corresponding OOD score maps produced by PEBAL and their histograms. By comparison, we observe a significant dampening in the OOD prediction score due to the domain shift, as the unknown object outlined in black lines becomes less distinguishable in the OOD score map of the second row, and the separation in the histogram between known and unknown classes diminishes.
>
> - In Figure 1(b), we show the qualitative results of different methods tested on the original FS Static dataset and its domain-shift version. We quantify the drop in PEBAL's performance with the added domain shift and the substantial mitigation of this deterioration when combined with our method. We also examine test-time adaptation methods like TBN and Tent, which gain improvement in high-domain-shift datasets at the cost of the performance in the original dataset. Unlike these, our method enhances performance across both non-domain-shift and domain-shift scenarios, reflecting greater robustness to real-world settings where test data can come from seen or unseen domains.
>
> ## W3: Implementation details of Tent on PEBAL
>
> We use the pretrained model and inference code of PEBAL provided by their official github repository and optimize the parameters of this pretrained model during test time by implementing Tent specifically.
>
> As detailed in Appendix B of our manuscript, we closely followed the original Tent paper and its official code to implement this combination. This involves using transductive batch normalization, calculating the entropy loss on the inlier classifier, and updating the affine parameters of all Batch Normalization layers. We also adopt the episodic training manner, a batch size of 1, and Adam optimizer, a setting consistent with both the Tent paper's guidance regarding their segmentation experiment and our own method's configuration. To ensure a fair comparison, we update the model only once for each test image and tune the learning rate on the FS Static-C dataset. These steps allow us to align our experimental setup with that of the Tent method and facilitate a meaningful comparison.
>
> ## W4: Our Method Combined with Various OOD Detection Methods on FS Static-C
>
> Thank you for the suggestion. As requested, we extended our evaluation to include not only PEBAL but also the remaining five OOD detection methods. The results in Table B1 below demonstrate that our method consistently enhances the robustness of these OOD detection techniques in scenarios with potential domain shifts. Specifically, we achieved significant performance gains on the FS Static-C dataset for all OOD detection methods, with an average increase of AUC of around 20% (cf. row #2), AP of 10% (cf. row #4), and a decrease of FPR95 of nearly 80% (cf. last row). Some results on the FS Static-C dataset are even better than the original performance of the corresponding methods on the non-domain-shift FS-Static dataset, such as MSP, Entropy, and Meta-OOD, as shown in the table. By combining our method with various OOD detection methods, we demonstrate its efficacy and general applicability across different scenarios.
>
> Table B1: We display additional results of our method combined with various previously established OOD detection methods on both the FS Static and FS Static-C datasets.
>
> | Metric | Dataset     | MSP   | + Ours    | Entropy | + Ours    | Max logit | + Ours    | Energy | + Ours    | Meta-OOD | + Ours    |
> |--------|-------------|-------|-----------|---------|-----------|-----------|-----------|--------|-----------|----------|-----------|
> | AUC $\uparrow$    | FS-Static   | 92.36 | **93.91** | 93.14   | **95.18** | 95.66     | 95.48     | 95.90  | **96.00** | 97.56    | **98.19** |
> |        | FS-Static-C | 70.85 | **92.97** | 71.23   | **94.33** | 74.13     | **94.80** | 74.02  | **95.41** | 78.34    | **98.06** |
> | AP $\uparrow$     | FS-Static   | 19.09 | **26.57** | 26.77   | **39.57** | 38.64     | **41.23** | 41.68  | **41.84** | 72.91    | **83.11** |
> |        | FS-Static-C | 10.52 | **20.81** | 14.32   | **30.78** | 23.60     | **31.13** | 22.36  | **32.13** | 52.31    | **75.75** |
> | FPR95 $\downarrow$  | FS-Static   | 23.99 | **20.80** | 23.31   | **18.98** | 18.26     | 20.89     | 17.78  | **17.63** | 13.57    | **11.63** |
> |        | FS-Static-C | 100.0 | **22.58** | 100.00  | **20.21** | 89.94     | **23.59** | 89.94  | **18.63** | 100.0    | **11.17** |

---

> ### Comment · Reviewer_TGMY · 2023-08-12
>
> While the rebuttal addresses several points of the reviews, the changes and additional experiments discussed are several. The paper should go through another round of peer review with many changes. Therefore, I will not increase my score.

---

> > ### Author Response · Authors · 2023-08-13
> > **Response to Reviewer TGMY**
> >
> > Dear Reviewer TGMY,
> >
> > Thank you for your thoughtful feedback. Regarding the extent of the changes, we'd like to emphasize that our rebuttals are targeted responses to the comments raised by reviewers. Rather than altering the main content of our paper, they are intended to complement and reinforce our existing arguments. Therefore, those additional materials will primarily be placed in the appendix and should not affect the paper's main structure.
> >
> > Here is a summary of our additional results and changes:
> >
> > **Summary of the Additional Results:**
> > - **SMIYC Benchmark (General Response 1):** These results enhance our testing on the domain-shift dataset, aligning with existing findings on FS Static -C and the Road Anomaly datasets.
> > - **Lost and Found Dataset / online FS test set (Response to Reviewer Nknk):** These complementary results align with existing findings on the Fishyscapes Lost&Found and Fishyscapes Static offline datasets.
> > -  **Combination with Other OOD Detection Methods on FS Static -C (Response to Reviewer TGMY):** The results demonstrate the general applicability of our method which is consistent with our existing findings shown in Table 2.
> > - **Overhead Analysis (General Response 2):** Additional inference time has been discussed as a potential limitation in our conclusion part.  While efficiency is not our paper's focus, we have included these results to address concerns.
> > -  **Combination with Architectures without BN (Response to Reviewer XNCt):** Our paper primarily focuses on architectures with BN. The experiments with Segmenter (using Layer Norm) demonstrate extendibility to other architectures but are not a central focus of our paper.
> >
> > **Summary of Other Changes:**
> > - **The Detailed Figure 2:** This enhanced version is already provided in the attached PDF. It can directly be replaced with the original Figure 2, with minimal impact on other parts.
> > - **Notation Clarity, Minor Typos, Detailed Caption, Formula Punctuated (Response to Reviewer TGMY, 7M1E):** These are minor points and will not change the overall content.
> >
> > As outlined above, the additional results and changes are designed to support and reinforce our paper without altering its main content. We appreciate your thoughtful insights and remain open to any further suggestions.
> >
> > Sincerely,
> > Authors

---

### Official Review · Reviewer_Nknk · 2023-07-07

**Soundness:** 3 good
**Presentation:** 4 excellent
**Contribution:** 4 excellent
**Rating:** 5
**Confidence:** 4

**Summary:**

The proposed method considers the OOD sample with domain shift and semantic shift.  They address the problem of current OOD approaches that often ignore to handling domain shifts. They introduce an anomaly-aware test-time adaptation method that
jointly tackles domain and semantic shifts. The experiments on different benchmarks, the proposed method demonstrated significant performance gains on various OOD segmentation benchmarks. They show significantly robust performance on benchmarks with notable domain shifts.

**Strengths:**

The paper has well-organized writing and clear motivation for each part of the proposed method.

The proposed method is relatively novel and presented clearly.

The proposed method achieves SOTA results with large performance gains compared with other SOTA methods.

The proposed method first time focuses on handling domain shift OOD detection, which has been overlooked by previous OOD methods.



**Weaknesses:**

From Figure 2, readers cannot figure out any details regarding each of the proposed components and the overall framework. I would suggest the authors could include more details in the figure to have more detailed basic information for each of their contributions.

Although the authors compare their approach to the Road Anomaly dataset for domain shift effectiveness evaluation. The domain shift is even more significant in SegmentIfYouCan benchmarks. Hence, I would suggest the authors to show more results on the SegmentIfYouCan benchmark and see if the proposed ATTA can improve the results. From what I observed, the performance of previous SOTA such as PEBAL, and Meta-OOD, perform relatively worse on the domain-shifted benchmarks like SegmentIfYouCan.

It would be better if the authors can compare their approach to the online Fishyscape testing set. The validation set only contains a few images, which may not be enough to effectively quantify the performances of the method.

Computational efficiency is important to perform real-time detection for self-driving systems, could authors present the training and inference time of the proposed approach and compare it with other methods?



**Questions:**

The computation efficiency of the proposed method is unknow.

**Limitations:**

According to the paper, the model's improvement is less noticeable with weaker backbones. The proposed method will add additional time.

---

> ### Author Rebuttal · Authors · 2023-08-10
>
> Thank you for recognizing our contribution and for your thoughtful and detailed feedback.  We have presented our results of SMIYC and analyzed the inference time of our method in the general response. Please find our responses to your other concerns below:
> ## W1: Detailed Figure 2 - Model Overview
>
> We thank the reviewer for the suggestion and have provided a revised Figure 2 in the attached PDF, containing details of our model design. This updated figure provides a more detailed illustration of each of the proposed components and the overall framework.
>
> ## W3: Results for online FS Test set
>
> We thank the reviewer for the suggestion and have submitted our results to the online Fisyscapes testing server during rebuttal. Unfortunately, due to an unexpected shutdown of the testing server, we are unable to provide the results at this moment. We promise to provide the results when they are available and will include them in our revised paper.
>
> To evaluate our method on a larger benchmark, we have tested our model on the Lost And Found dataset [1], which contains 1203 images. As presented in Table 1, our method outperforms the previous OOD detection methods, Max Logit and PEBAL across all metrics, validating our approach's efficacy. We note that since this dataset contains minimal domain shift, our improvement is relatively small compared to the performance gains observed on other datasets exhibiting more pronounced domain shifts.
>
> Table A1. Comparison of methods on the Lost And Found dataset. Results were obtained by running the publicly available pre-trained models on our device.
>
> |  Lost And Found  |           |           |           |
> |----------------|:---------:|:---------:|:---------:|
> | Methods          | AUROC$\uparrow$    | AP$\uparrow$       | FPR95$\downarrow$    |
> | Max Logit        | 92.73     | 53.22     | 52.51     |
> | Max Logit + ATTA | **93.66** | **57.93** | **47.27** |
> | PEBAL            | 96.88     | 71.21     | 14.63     |
> | PEBAL + ATTA     | **96.95** | **72.39** | **14.55** |
>
>
> [1]  Peter Pinggera et al. Lost and found: detecting small road hazards for self-driving vehicles. IROS, 2016.
>
> ## W4 & Question: Computational Efficiency
>
> We appreciate the reviewer's concern about computational efficiency.
> - **Training Time:** Our method is designed primarily for test time and can leverage a pretrained model, eliminating additional training. If a pretrained model is not available, we can follow the standard closed-world segmentation training procedure, utilizing the training set. The training time would then depend on the specific segmentation model used, but our method does not add to this time.
> - **Inference Time:** In General Response 2, we demonstrate the inference time of our method and compare it with other methods. We conclude that our method takes only 1.25-2.25 times the pure inference time and is faster than Tent and some OOD detection methods with posthoc operations, such as ODIN, Synboost, and Mahalanobis Distance.
>
> In case the reviewer specifically means the training and inference time within our self-training procedure, we provide a detailed breakdown in Table A2. We note that the inference time to get the final OOD score is only 0.1s since we only need to do the re-calculation for the classification block.
>
> Table A2:  Detailed time (second per image) for our self-training procedure. The experimental setting is kept the same as in Table G2. The initial forward includes the forward pass of both the feature extractor and classifier.
> | Components | Initial Forward | Loss and Backward | Classifier Forward |
> |:----------:|:---------------:|:-----------------:|:------------------:|
> |  Time (s)  |       1.2       |        0.2        |         0.1        |

---

> > ### Author Response · Authors · 2023-08-21
> > **Online Fishyscapes Testing Set Results**
> >
> > Dear Reviewer Nknk,
> >
> > We have now obtained our results for the online Fishyscapes testing set. As presented in Table A3, our method outperforms the previous state-of-the-art, PEBAL. We appreciate your insights and remain open to further suggestions.
> >
> > Table A3: Results on the Fishyscapes online test benchmark. The results for our model were obtained by submitting it to the benchmark organizer. The results for PEBAL were taken from their published paper.
> >
> > |              | Online FS Lost & Found |      |
> > |--------------|:---------------:|:----:|
> > |              |        AP$\uparrow$       |  FPR$\downarrow$  |
> > | PEBAL        |      44.17      | 7.58 |
> > | PEBAL + ATTA (Ours) |      55.94      | 4.66 |
> >
> > |              | Online FS Static |      |
> > |--------------|-----------|------|
> > |              | AP$\uparrow$         | FPR$\downarrow$    |
> > | PEBAL        | 92.38     | 1.73 |
> > | PEBAL + ATTA (Ours) | 94.68     | 0.68 |

---

### Author Rebuttal · Authors · 2023-08-10

We thank all the reviewers for their time and constructive comments. In the following, we address some shared concerns in this general response and answer each individual question by replying to each reviewer. We also include an additional PDF containing a revised Figure 2, some examples showing the specific "domain shift" of the Road Anomaly datasets, and visualization results on the SMIYC validation set.

## 1. Results on the SMIYC benchmark

We thank the reviewers for the suggestion of evaluating our method on the SegmentMeIfYouCan (SMIYC) [1] benchmark, which particularly includes significant domain shifts such as variations in illumination and weather. We submit our outputs to the benchmark test set and present the results in Table G1. Our experiments on the RoadAnomaly21 and RoadObstacle21 datasets, both part of the SMIYC benchmark, demonstrate significant improvements over the previous SOTA method PEBAL [2]. Notably, in the RoadObstacle21 dataset, PEBAL's performance is hampered by a lack of robustness, whereas our method has increased the AUPRC score from 5.0% to 76.5%. This validates our method's adaptability to substantial domain shifts. We also present the visualization results on the validation sets in the attached PDF.

Table G1: Results on the SMIYC official test benchmark. The results for our model were obtained by submitting the model outputs to the benchmark organizer, as required. The results for PEBAL were taken from the benchmark's official website.

| RoadAnomaly21              |  |        |          |      |          |
|--------------|:--------------:|:------:|:--------:|:----:|:--------:|
| Methods             | AUPRC $\uparrow$         | FPR95$\downarrow$ | sIoU gt $\uparrow$ | PPV $\uparrow$ | mean F1 $\uparrow$|
| PEBAL        | 49.1           | 40.8   | 38.9     | 27.2 | 14.5     |
| PEBAL + ATTA  | **67.0**       | **31.6** | **44.6** | **29.6** | **20.6** |

| RoadObstacle21              | |        |          |      |          |
|--------------|:--------------:|:------:|:--------:|:----:|:--------:|
| Methods             | AUPRC $\uparrow$         | FPR95$\downarrow$ | sIoU gt $\uparrow$ | PPV $\uparrow$ | mean F1 $\uparrow$|
| PEBAL        | 5.0            | 12.7   | 29.9     | 7.6  | 5.5      |
| PEBAL + ATTA  | **76.5**       | **2.8**  | **43.9** | **37.7** | **36.6** |

[1] Chan Robin, et al. Segmentmeifyoucan: A benchmark for anomaly segmentation. NeurIPS, 2021.
[2] Yu Tian, et al. Pixel-wise energy-biased abstention learning for anomaly segmentation on complex urban driving scenes. ECCV, 2022.

## 2. Inference Time
We acknowledge the reviewers' concern about computational efficiency.
- In this paper, we address a general dense out-of-distribution detection problem, where our method may be applied to various applications beyond self-driving, and thus real-time performance is not the primary focus.
- In response to the reviewers' request, we have evaluated the average inference time for each image (see Table G2), revealing that our method is only 2.25 times slower than direct inference, and faster than another test-time adaptation method, Tent [1], and some ood detection methods with posthoc operations: ODIN [2],  Mahalanobis Distance [3] and Synboost [4].
- This efficiency is attributed to our design, which updates only once per image and confines learnable parameters to the classifier block (cf. lines 238, 278 of the manuscript). The latter design enables us to perform backward and the subsequent forward pass only on the classifier block, and is the main reason we achieve much faster inference than Tent.
- Furthermore, in scenarios where data from the same domain are known, we can reduce the computation by performing domain-shift detection only once, maintaining the variable for subsequent images. To illustrate this, we show the inference time without domain-shift detection, which further reduces it to 1.25 times the direct inference speed.
- Practically, our episodic training model allows for parallel inference across multiple processors. With ongoing hardware advancements, we anticipate a further reduction in the time gap.

Table G2: Comparison of inference time (seconds per image). We calculate the complete time from input to the final prediction and/or ood score (See [5] for details). Experiments are conducted on one NVIDIA TITAN Xp device, and results are averaged over all images in the FS Lost & Found validation set, with image size (1024 x 2048).
|  Methods  | Direct Inference | ATTA (Ours) | ATTA (Ours) w/o SBN | Tent | ODIN | SynBoost | Mahalanobis |
|:---------:|:----------------:|:-----------:|:-------------------:|:----:|:----:|:--------:|:-----------:|
|  Time (s) |        1.2       |     2.7     |         1.5         |  5.1 |  9.2 |    3.0   |    224.2    |

[1] Dequan Wang, et al. Tent: Fully test-time adaptation by minimization. ICLR, 2020.
[2] Shiyu Liang, et al. Enhancing The Reliability of Out-of-distribution Image Detection in Neural Networks. ICLR, 2018.
[3]  Kimin Lee, et al. A Simple Unified Framework for Detecting Out-of-Distribution Samples and Adversarial Attacks. NeurIPS, 2018.
[4] Giancarlo Di Biase, et al. Pixel-Wise Anomaly Detection in Complex Driving Scenes. CVPR, 2021.
[5] https://deci.ai/blog/measure-inference-time-deep-neural-networks/.

---

> ### Comment · Area_Chair_ahms · 2023-08-18
> **Authors rebuttal**
>
> I acknowledge the authors rebuttal and I am encouraging reviewers to reflect on.

---

### Decision · Program_Chairs · 2023-09-21

**Decision:**

Accept (poster)

**Comment:**

The paper tackled the problem of OOD detection for semantic segmentation in the presence of domain shift. The paper tackles an interesting aspect of ML safety where TTA and OOD are nicely combined. Reviewers have raised many questions and the authors posted additional results, and clarifications. All reviewers concerns have been addressed, however 2 reviewers suggested that the paper should go through another round of peer review given the modifications made. To this AC, the additional results are mainly clarifications of some aspects and evaluation on more dataset, and hence the core contribution is still valid. I suggest that the authors incorporate all the extra analysis and results in main paper and appendix.